# Androgen receptor is a determinant of melanoma targeted drug resistance

Anastasia Samarkina [1], Markus Kirolos Youssef [1], Paola Ostano [2], Soumitra Ghosh[3], Min Ma[1], Beatrice Tassone[1], Tatiana Proust [1], Giovanna Chiorino [2], Mitchell P. Levesque[4], Sandro Goruppi [5] & Gian Paolo Dotto[1,3,5,6] ✉

Melanoma provides a primary benchmark for targeted drug therapy. Most melanomas with BRAF[V600] mutations regress in response to BRAF/MEK inhibitors (BRAFi/MEKi). However, nearly all relapse within the first two years, and there is a connection between BRAFi/MEKi-resistance and poor response to immune checkpoint therapy. We reported that androgen receptor (AR) activity is required for melanoma cell proliferation and tumorigenesis. We show here that AR expression is markedly increased in BRAFi-resistant melanoma cells, and in sensitive cells soon after BRAFi exposure. Increased AR expression is sufficient to render melanoma cells BRAFi-resistant, eliciting transcriptional changes of BRAFi-resistant subpopulations, including elevated *EGFR* and *SERPINE1* expression, of likely clinical significance. Inhibition of *AR* expression or activity blunts changes in gene expression and suppresses proliferation and tumorigenesis of BRAFi-resistant melanoma cells, promoting clusters of CD8[+] T cells infiltration and cancer cells killing. Our findings point to targeting AR as possible co-therapeutical approach in melanoma treatment.

Significant differences exist in melanoma mortality between men and women across all ages after adjusting for tumor variables (Breslow thickness, histologic subtypes, body site, and metastatic status)[1]. As for sexual dimorphism in other cancer types[2], even for melanoma, differences in sex hormone levels and/or downstream pathways are likely to play a role[3]. Sex hormone signaling can affect cancer susceptibility through multiple intrinsic and extrinsic mechanisms, impacting cancer stem cell renewal, the tumor microenvironment, the immune system, and the metabolic balance of the organism[2,4–6]. As early as 1980, it was proposed that differences in androgen levels could help explain the lower survival of male versus female melanoma patients[7]. Recent epidemiological evidence links elevated free testosterone levels in male human populations with a high risk of melanoma as the only other cancer type besides prostate[8].

In our recent work, we have found that the androgen receptor (AR) gene is heterogeneously expressed in melanoma cells, both at the single-cell intralesional level and among lesions at various stages of the disease[9]. Irrespective of expression levels, silencing of the *AR* gene and pharmacological inhibition of AR activity suppresses proliferation and induces cellular senescence of a relatively large panel of melanoma cells from both male and female patients[9]. AR plays an essential function in this context by bridging the transcription and DNA repair machinery, maintaining genome integrity. In both cultured melanoma cells and tumors in vivo, *AR* gene silencing or treatment with AR inhibitors leads to chromosomal DNA breakage in the absence of other exogenous triggers, leakage into the cytoplasm, STING activation, and a STING-dependent pro-inflammatory cascade[9].

[1]Department of Immunobiology, University of Lausanne, Épalinges, Switzerland. [2]Cancer Genomics Laboratory, Edo and Elvo Tempia Valenta Foundation, Biella, Italy. [3]ORL service and Personalized Cancer Prevention Program, Centre Hospitalier Universitaire Vaudois, Lausanne, Switzerland. [4]Department of Dermatology, University Hospital Zürich, University of Zürich, Zürich, Switzerland. [5]Cutaneous Biology Research Center, Massachusetts General Hospital and Department of Dermatology, Harvard Medical School, Charlestown, MA, USA. [6]International Cancer Prevention Institute, Épalinges, Switzerland. ✉e-mail: gdotto@mgh.harvard.edu

In the present study, we assessed the translational significance of suppressing AR signaling in the context of melanoma response to targeted drug treatments, specifically BRAF inhibitors. -50% of all melanomas harbor BRAF[V600] mutations, with >90% of these expressing the V600E or K amino acid substitution. Although >80% of patients with BRAF[V600E/K] melanomas initially respond to highly specific BRAF and MEK inhibitors (BRAFi/MEKi), nearly all relapse between seven months to two years[10]. Most BRAFi/MEKi-resistant melanomas are also resistant to immunotherapies[11], with a cancer cell-instructed mechanism that does not depend on selection by the immune system[12,13]. Initial treatment of melanoma patients with BRAFi/MEKi elicits recruitment and activation of immune cells[14], similar to what we found in mouse xenografts with melanoma cells with *AR* gene silencing or inhibition[9]. In melanomas with acquired BRAFi/MEKi resistance, an opposite modulation of the immune cell response occurs, which can be attributed, in part, to epigenetic/transcriptional regulatory changes that have the potential of being pharmacologically reversed[14].

We show that increased AR expression and activity are part of the response of melanoma cells with BRAF[V600] mutations to treatment with BRAF inhibitors and that increased AR expression is sufficient to render these cells resistant to these drugs, inducing transcriptional changes of BRAFi-resistant subpopulations. Conversely, treatment with AR inhibitors suppresses the proliferation and tumorigenicity of BRAFi-resistant melanoma cells, enhancing CD8[+] T cell infiltration. The findings align with a comprehensive series of clinical and mouse model data published while this paper was under review[15], raising the prospect that targeting AR, a standard treatment for metastatic prostate cancer, could also be used to augment the effectiveness of targeted therapy for melanoma.

## Results

### Acquisition of Dabrafenib resistance is associated with increased AR expression and activity

Acquisition of BRAFi resistance by melanoma cells can be a dynamic process that is induced in culture by the drug treatment[16,17]. Treatment of a panel of primary and established human melanoma cells with multistep increases of the BRAF inhibitor Dabrafenib (DAB), utilizing similar concentrations as in previous studies[18,19], resulted in the emergence of cells with greater capability to proliferate in the presence of this compound (Supplementary Fig. 1a–h). RT-qPCR and immunoblot analysis showed substantially increased AR expression, already at lower doses of DAB treatment (Fig. 1a). A consistent increase in AR expression was found in additional primary and established melanoma cells selected for BRAFi resistance by immunoblot and immunofluorescence analysis (IF) as well as RT-qPCR (Fig. 1b, c, Supplementary Fig. 1a-c, Supplementary Fig. 1f-m), with variable changes in MAPK expression and activation, consistent with previous publications[20] (Supplementary Fig. 1n). While AR expression was upregulated in all BRAFi-resistant cell lines relative to parental cells, other genes connected with the acquisition of BRAFi resistance, such as *MITF, SOX9, SOX10, ZEB1,* and *ZEB2*[21] were more unevenly modulated (Fig. 1d; Supplementary Fig. 1f-m).

Work was extended by global transcriptomic analysis of five melanoma cell lines selected for BRAFi resistance versus parental controls. 351 genes were found to be significantly differentially expressed (absolute $\log_2$ FC >1, $p < 0.05$) in all cell lines in concomitance with BRAFi resistance, with *AR* among the top up-regulated genes (Fig. 1e, Supplementary Data 1). Gene set enrichment analysis (GSEA) of the combined profiles of the BRAFi-resistant versus parental cell lines showed positive enrichment of an established AR-responsive gene signature (Wikipathways[22]: https://www.gsea-msigdb.org/gsea/msigdb/) and gene signatures previously connected with acquisition of BRAFi resistance, specifically epithelial-mesenchymal transition (EMT) and undifferentiated and neural crest melanoma cells (UNDIF and UNDIF-NC)[23], as well as EGFR[24] and TGF-ß signaling[24] (Fig. 1f, g).

To further assess the clinical significance of the findings, we evaluated AR expression by IF analysis of matched lesions excised from the same patients before and after BRAFi/MEKi therapy. Upregulation of AR expression was also found in the clinical setting (Fig. 1h).

Thus, the acquisition of BRAFi resistance in multiple melanoma cell lines is consistently linked with increased AR expression and activity.

### BRAFi treatment induces short-term AR expression in melanoma cells through an AR positive feedback loop

AR upregulation may result from chronic BRAFi treatment or be part of an acute response. We found that pronounced induction of *AR* expression occurred in a panel of primary and established melanoma cells already by 48 hours of treatment with DAB and other BRAF and MEK inhibitors (Fig. 2a), while inhibitors of other key signaling pathways, such as NF-κB, STAT3, and AP-1 exerted no such effect (Supplementary Fig. 2a).

Immunoblot analysis showed that AR expression was increased in various melanoma cell lines upon 48-h, 72-h, and one-week-long DAB treatment (Fig. 2b-d, Supplementary Fig. 2b). For further mechanistic insights, we focused on A375 melanoma cells. A detailed time course analysis confirmed induction of *AR* mRNA levels after 48 hours of DAB treatment with further upregulation after 72 hours (Fig. 2e). IF analysis revealed an increase in nuclear AR protein levels already after two hours of DAB treatment with a more pronounced increase by 24 hours and 48 hours (Fig. 2f; Supplementary Fig 2c). Cell fractionation and immunoblot analysis confirmed enhanced nuclear localization of the AR protein by 48 hours of DAB treatment similar to that found in BRAFi-resistant cells (Fig. 2g, h; Supplementary Fig 2d, e).

These findings raised the exciting possibility that AR binds to chromatin already at early times of DAB treatment and upregulates genes of interest, including itself. Chromatin immunoprecipitation-sequencing (ChIP-seq) analysis using anti-AR antibodies showed a marked increase in AR binding to genomic regions encompassing the transcription start sites (TSS) in A375 melanoma cells after 48 hours of DAB treatment (Fig. 3a).

To complement these findings, we performed transcriptomic analysis of A375 cells and two other melanoma lines plus/minus DAB treatment for 48 hours. 362 genes were consistently induced in all three lines (absolute $\log_2$ FC >1, FDR < 0.05; Supplementary Data 2). Out of these, 123 genes are direct AR targets (Supplementary Data 2). The gene ontology analysis revealed enrichment of these gene families associated with ERK/MAPK signaling (Fig. 3b). Importantly, the *AR* gene was among the top five target genes bound by AR in DAB-treated A375 melanoma cells (Fig. 3c). A major peak of AR binding induced by DAB treatment was found overlapping the TSS site of the AR gene, with three other DAB-induced peaks being identified at exon 3 and 5 −9 regions of the gene (Fig. 3d, Supplementary Fig. 3a).

To assess whether the identified AR-binding peaks in the DAB-treated melanoma cells reflect the general binding of AR to its own gene, we examined publicly available ChIP-Seq profiles, using the Cistrome DB toolkit (http://dbtoolkit.cistrome.org/). This analysis showed that the top transcription factors binding on AR peaks in the *AR* gene were AR itself, with other transcription factors including the pioneering factor FOXA1, which has been shown to co-function with AR[25], as well as the Estrogen Receptor and the Progesterone Receptor (Fig. 3e, Supplementary Fig. 3b).

To functionally test whether AR itself is involved in AR upregulation in response to DAB treatment, we co-administered two proteolysis-targeting chimeras (PROTACs), ARCC4[26] and ARV110[27], with DAB. Concomitant PROTAC treatment suppressed DAB-induced AR mRNA expression, while no such inhibition was established by the co-administration of EGFR and TGF-β inhibitors (Afatinib and SD-208, respectively). Interestingly, inhibiting AP1, which is known to synergize with AR in the control of gene expression[28], also suppressed the DAB-

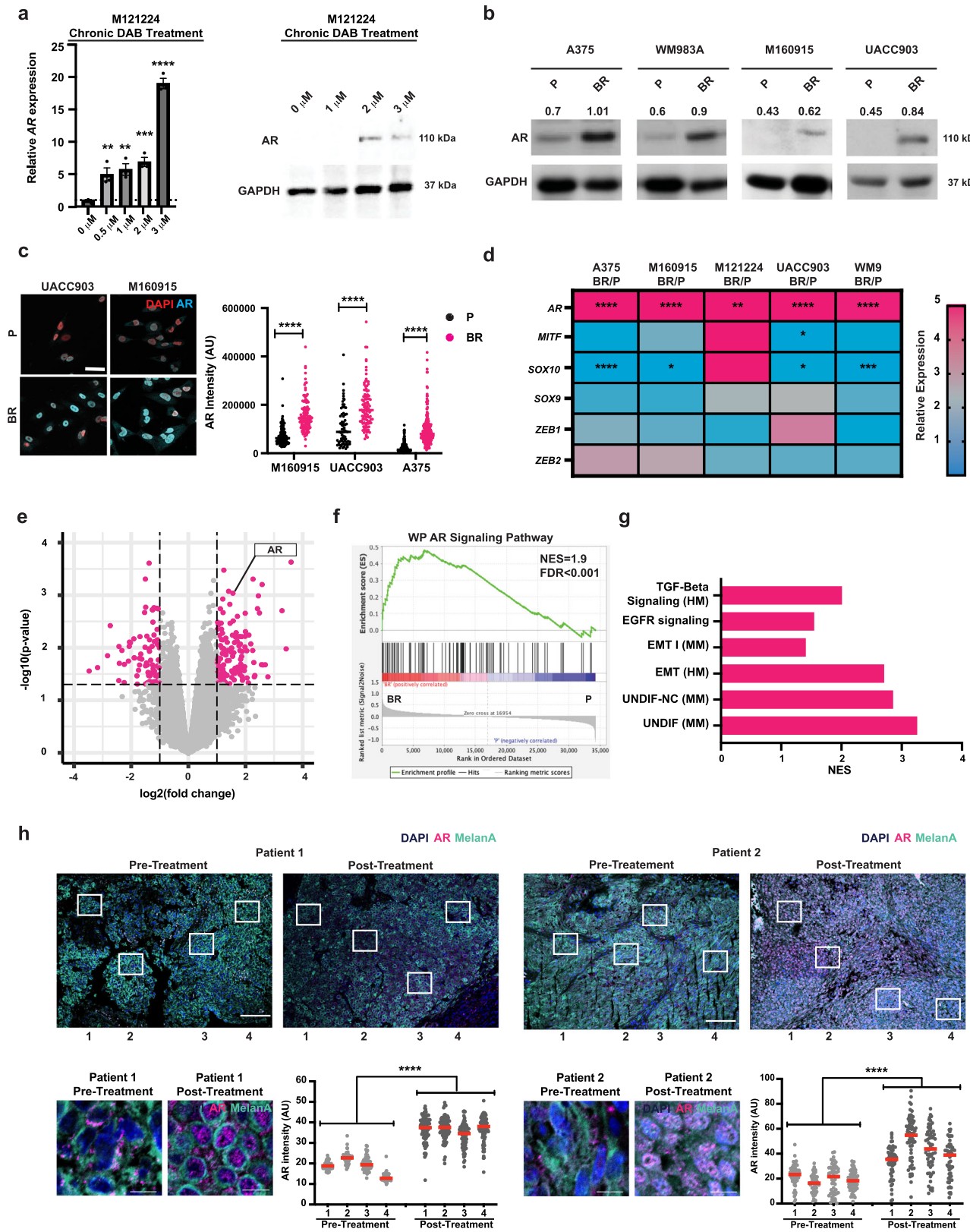

induced expression of AR (Fig. 3f). IF analysis confirmed the suppression of DAB-induced AR expression by ARCC4 and the AP-1 inhibitor T-5224 after 48 h (Fig. 3g).

Thus, *AR* gene expression is consistently induced in BRAFi-resistant melanoma cells as well as in naïve melanoma cells upon acute exposure to BRAF/MEK inhibitors, involving AR nuclear translocation and a positive feedback loop.

## Increased AR expression triggers a BRAFi-resistant phenotype

To assess the functional significance of these findings, we infected three different melanoma lines (A375, WM9, and M14) with an AR overexpressing lentivirus (AR OE) versus LacZ-expressing control (CNTRL) (Supplementary Fig. 4). In dose-response cell growth assays, the half-maximal inhibitory concentration ($IC_{50}$) of DAB at 72 h of treatment was drastically increased by AR overexpression in all three

**Fig. 1 | Chronic BRAFi treatment of melanoma cells results in increased androgen receptor (AR) expression. a** RT-qPCR and immunoblot analysis (WB) of AR expression in primary human melanoma cells (M121224) with weekly increases of BRAF inhibitor Dabrafenib (DAB, 0.5, 1, 2, and 3 μM). RT-qPCR was normalized to *RPLPO* and WB to GAPDH. Mean ± SEM. n(biological replicates)= 3, unpaired, two-tailed t-test. **p < 0.01, ****p < 0.0001. Additional melanoma lines are in Supplementary Fig. 1a–c. **b** AR and GAPDH WBs of primary (M160915) and established melanoma cells (A375, WM983A, and UACC903), selected for BRAFi resistance (BR) as in **a**, versus parental cells (P). Relative AR intensity levels (numbers) were normalized to GAPDH. **c** AR immunofluorescence analysis (IF) with DAPI (red) in P and BR melanoma cells established as in **a**. Representative images and AR nuclear signal quantification per cell (arbitrary units). n(cells/sample) >100, unpaired two-tailed t-test, **** p < 0.0001. Scale bar: 40 μm. **d** Heatmap of *AR* expression, as assessed by RT-qPCR analysis with *RPLPO* normalization, in BR primary (M160915, M121224) and established melanoma cells (A375 and WM983A) versus parental cells (P). Up-(magenta) and down- (blue) regulated genes. Two-tailed multiple comparison t-

test, n(biological replicates)=3. Unpaired t-test. ***p < 0.001, ****p < 0.0001.
**e** Volcano plot of transcriptional changes consistently elicited in five BR versus P melanoma cells (A375, WM9, UACC903, M160915 and M121224). Fold change (log₂) and *p*-value (−log₁₀). Red dots show genes with a p < 0.05, two-tailed t-test, and fold-change > −1 and 1. The gene list is in Supplementary Data 1. **f** Gene set enrichment analysis (GSEA) of BR versus P melanoma cells transcriptional profiles using an *AR* gene signature from Wikipathways (WP)[53]. Black bars indicate the individual genes, enrichment is in green. Normalized enrichment score = NES. **g** GSEA and NES of BR versus P transcriptional profiles in five melanoma cells (same as **e**), using gene signatures from Hallmark (HM)[22], WP[53], biocarta (BC) and melanoma studies (MM)[23,48]. **h** AR (magenta) and MelanA (cyan) IF, with DAPI (blue), of matched pre- and post-BRAFi/MEKi treatment lesions (patients 1 (A) and 2 (B)). Representative low- and high-magnification images of areas quantified (boxes 1–4). AR signal per cell (arbitrary units), mean ± SD, *n*(cells/sample) >50, paired, two-tailed *t*-test, ****p < 0.0001. Scale bar: 100 μm and 10 μm, respectively.

cell lines (Fig. 4a). In one-week cell imaging assays (Incucyte), the proliferation of control A375 cells was suppressed by DAB treatment at all tested concentrations. In contrast, that of AR overexpressing cells was initially reduced, but cultures eventually attained the same density as untreated controls (Fig. 4b). In parallel, DAB treatment induced cell death to a much greater extent in control than AR overexpressing cells (Fig. 4c).

The findings were expanded by clonogenicity assays. DAB treatment reduced the number of colonies produced by control cells. AR overexpression enhanced the colony-forming ability of cells already under basal conditions and effectively counteracted the DAB-induced decrease in colony number (Fig. 4d). Similar protective effects were exerted by AR overexpression in A375 cells treated with DAB individually and in combination with the MEK inhibitor Trametinib (TRA) (Fig. 4e). The findings were complemented by 3D tumor spheroid invasion assays[29]. AR overexpression resulted in increased invasion capability of cells under basal conditions, which remained higher than controls upon DAB treatment (Supplementary Fig. 5).

Altogether, the findings indicate that AR overexpression in melanoma cells effectively counteracts growth suppression by BRAF inhibition.

## Increased AR expression perturbs the transcriptional response of melanoma cells to BRAFi

For further mechanistic insights, we undertook a global transcriptomic analysis of the three melanoma cell lines tested above. A large fraction of genes were similarly modulated in control and AR overexpressing cells at 48 hours of DAB treatment (Fig. 5a, Supplementary Data 2). Gene families related to cell cycle and DNA replication were commonly downmodulated, consistent with the decreased rate of proliferation that also occurred with AR overexpressing cells at early times of DAB exposure (Fig. 5b, Supplementary Data 2). By contrast, the mitochondrial pro-apoptotic pathway genes were upregulated by DAB treatment to a much greater extent in control than AR overexpressing cells, consistent with the differential pro-apoptotic effects (Fig. 5b, Supplementary Data 2). Gene families related to pro-inflammatory signaling pathways (interferon α/ß and TNF-α) were significantly induced by DAB treatment selectively in control cells. Conversely, genes of the EGFR and TGF-ß pathways involved in melanoma progression and targeted drug resistance[16,24] were induced by DAB treatment in AR overexpressing cells, to a much greater extent than in control cells (Fig. 5b, Supplementary Data 2).

The findings were expanded by Gene Set Enrichment Analysis (GSEA), showing that signatures of interferon α- and γ response were highly induced by DAB treatment of control cells (Fig. 5c), with a strong difference in the DAB-treated control versus AR overexpressing cells (Fig. 5d). Importantly, an antigen presentation gene signature

encompassing many major histocompatibility class I (MHC I) genes was also highly enriched in the DAB-treated control cells, with a profound difference relative to AR overexpressing cells (Fig. 5e, f).

Thus, increased AR expression in melanoma cells subverts the transcriptional response of melanoma cells to BRAF inhibition, with suppression of pro-apoptotic, immunomodulatory, and antigen presentation pathways and enhancement of pathways implicated in tumor progression and BRAFi resistance.

## Increased TGF-ß and EGFR signaling can account for DAB resistance resulting from increased AR expression

To identify genes or sets of genes that are permanently modulated by increased AR expression and may account for their long-term BRAFi resistance, we compared the transcriptional profiles of the three melanoma cell lines plus/minus AR overexpression under basal conditions. Next to the *AR* gene itself, *SERPINE1*, an established TGF-ß target with pro-tumorigenic functions[30,31], was the single most upregulated gene in all three AR overexpressing melanoma cells (Fig. 6a). Consistent with the transcriptomic results, RT-qPCR and immunoblot analysis showed a marked increase in *SERPINE1* (PAI-1) levels in all tested melanoma cell lines upon AR overexpression (Fig. 6b, c). Expression of the *SERPINE1* gene was also significantly upregulated in 4 out of 5 melanoma cell lines that were selected for DAB resistance (Supplementary Fig. 6), while it was not induced in parental cells at 48 h of DAB treatment (Supplementary Data 2).

For further mechanistic insights, we conducted ChIP-seq analysis using anti-AR antibodies on A375 melanoma cells with and without AR overexpression and compared the ChIP-seq profiles with those of AR in the same cells after early treatment with BRAFi (DAB). Our rationale was that short-term exposure of melanoma cells to BRAF inhibitors leads to the significant induction of AR expression, accompanied by gene expression changes that, however, are insufficient to cause BRAFi resistance. Initially, we focused on the *SERPINE1* gene that, as shown above, is strongly upregulated in melanoma cells overexpressing AR or selected for BRAFi resistance, but not following short-term BRAFi exposure. We observed significantly elevated AR binding peaks at the promoter, first exon, and second exon regions of *SERPINE1* in the AR-overexpressing cells relative to control cells or cells treated with DAB for 48 hours (Fig. 6d).

Through a global analysis of the ChIP-seq profiles, we identified a notable increase in AR binding near the transcription start sites (TSS) of several genes in both AR-overexpressing A375 melanoma cells and the same cells treated with DAB for 48 h. However, there was only partial overlap between AR target genes that were upregulated in these two conditions. One group of genes (65) was bound by AR and induced in both AR-overexpressing and DAB-treated cells, while another group (202) was selectively bound by AR and induced solely in the AR-overexpressing cells (Fig. 6e).

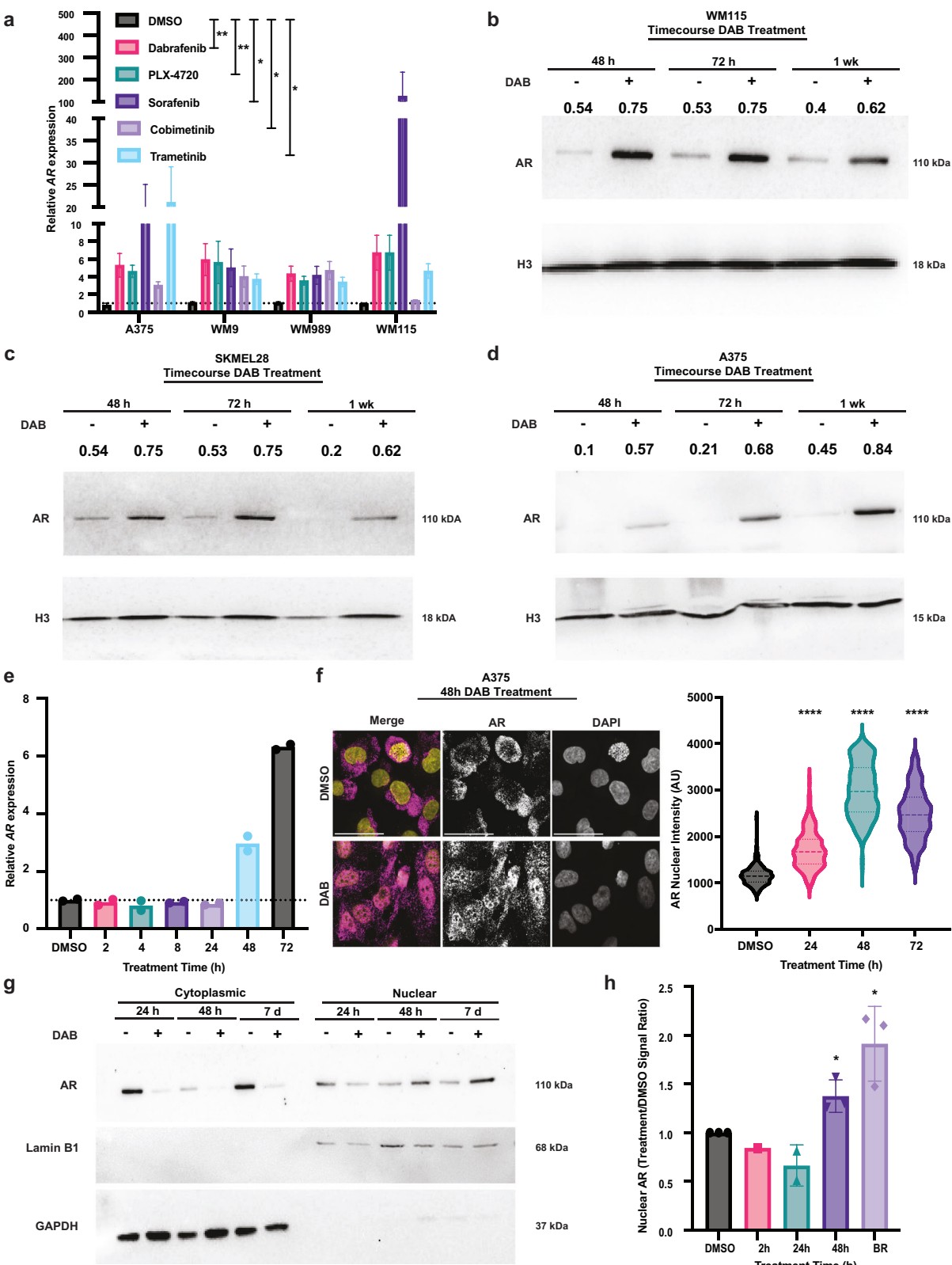

To identify known transcription factors and chromatin regulators that bind to sites overlapping with selective AR binding in AR-overexpressing cells, and potentially collaborate with AR function, we performed an unbiased analysis of publicly available ChIP-seq profiles using the Cistrome Data Browser and toolkit (http://dbtoolkit. cistrome.org/)[32]. Among the top-ranked transcription factors with a high score of overlap with the identified AR binding peaks across

various cell types were c-Myc, E2F1, and SP1. Additionally, BRD4, a protein involved in chromatin organization and transcription, along with a histone demethylase (KDB2B), a chromodomain helicase (CHD1), and an RNA polymerase subunit (POLR2A), were among the top-ranked proteins (Fig. 6f).

Gene ontology analysis of genes selectively bound by AR and upregulated in AR-overexpressing cells revealed a significant

**Fig. 2 | Acute BRAFi treatment of melanoma cells results in increased AR expression. a** RT-qPCR analysis of *AR* expression in melanoma cells at 48 hours of treatment with various kinase inhibitors targeting BRAF, MEK and other kinases of Ras-MAPK family (DAB, PLX-4720, Sorafenib: 0.5 μM; Cobimetinib, Trametinib, 5 nM), versus DMSO. Arbitrary units relative to untreated controls normalized to *RPLPO*. Mean ± SEM. n(biological replicates)=3, unpaired, two-tailed t-test. *p = 0.0286, **p = 0.0049. **b–d** AR and Histone3 (H3) WBs of WM115 **b** SKMEL28 **c**, and A375 **d** cells treated with DAB (0.5 μM) versus DMSO at the indicated time points. AR signal intensity (numbers) was normalized to H3. **e** RT-qPCR analysis of AR expression in A375 melanoma cells treated with DAB (0.5 μM) versus DMSO at the indicated time points. Arbitrary units relative to untreated controls, after *RPLPO* normalization. Mean ± SD, n(biological replicates)= 2. **f** AR IF in A375 melanoma cells treated with DAB (0.5 μM) versus DMSO control at the indicated time points. Representative images (left) and quantification (right) of AR nuclear signal per individual cells (arbitrary units), n(cell/sample)>200 cells, unpaired, two-tailed t-test, ****p < 0.0001. Color scale: yellow, DAPI; magenta, AR. Scale bar: 10 μm. **g** WB of AR subcellular distribution in A375 melanoma treated for 24 and 48 h and 7 days with DAB (0.5 μM) versus DMSO control cells. Shown are cytoplasmic and nuclear cell fractions, with LaminB1 as nuclear and GAPDH as cytoplasmic markers. **h** quantification of AR WB analysis of nuclear fractions of DAB-treated A375 melanoma cells versus DMSO at the indicated time points. Mean ± SD, n(biological replicates DMSO = 3; 2 h =1; 24 h = 2; 48 h = 3; BR = 3), 48 h and BR: unpaired two-tailed t-test, *p < 0.05.

enrichment of genes related to the Keap1-Nrf2, receptor tyrosine kinases and TGF-ß signaling pathways (Fig. 6g). Broader gene set enrichment analysis (GSEA) of the combined transcriptomic profiles of all examined melanoma cell lines, revealed a selective enrichment of genes related to TGF-ß and EGFR signaling in those overexpressing AR or selected with BRAFi resistance (Fig. 6h, Supplementary Fig. 6). The *EGFR* gene was itself also upregulated as a consequence of AR over-expression (Supplementary Fig. 6b, c).

To assess whether increased TGF-ß and/or EGFR signaling could account for the DAB resistance resulting from persistent AR expression, we treated cells with inhibitors targeting these pathways individually and in combination. Treatment with TGF-ß and/or EGFR inhibitors did not affect the proliferation of A375 parental cells, nor did it enhance the suppressive effects of DAB treatment (Fig. 6i). Similarly, the proliferation of AR-overexpressing cells was unaffected by treatment with TGF-ß and/or EGFR inhibitors alone. However, when these inhibitors were administered concomitantly with DAB, the sustained proliferation of AR-overexpressing cells was significantly reduced, with stronger suppressive effects observed when the two inhibitors were combined (Fig. 6i).

Thus, sustained AR expression results in differential AR targeting of genes with an increase of TGFß- and EGFR-related signaling that can be effectively targeted to suppress AR-induced BRAFi resistance.

### Increased AR expression elicits transcriptional changes of clinical significance found in BRAFi-resistant subpopulations

A recent study of the BRAFi response at the single-cell level in mouse Patients Derived Xenografts (PDXs) pointed to a transition of drug-naive melanoma cells to a BRAFi-induced starved-like (SMC) subpopulation branching out to three phenotypes[21]. By probing into the profiles of these distinct subpopulations, we found a highly enriched AR signature score in a specific BRAFi-tolerant subpopulation with elevated AXL expression and invasive features[21] (Fig. 7a). This same population was also found to have a positive enrichment score for the EGFR and TGF-ß gene signatures as well as for *SERPINE1* expression (Fig. 7a). Consistent with these findings, analysis of a single-cell dataset of melanoma cells obtained from patients prior to BRAFi treatment[33] showed a high AR signature score in tumor cell populations characterized by an "elevated AXL program" and low scores in other cell populations (Supplementary Fig. 6d).

These findings were extended by analyzing the composite transcription profiles of human melanoma cell lines and patients' tumors that cluster into four main groups along a two-dimensional differentiation trajectory[23]. Expression levels of the *AR* gene itself were positively associated with those of the *EGFR* gene in the most undifferentiated *AXL*-positive group connected with the targeted drug resistance[23] (Fig. 7b).

To assess the clinical significance of the results, we analyzed the transcriptomic profiles of melanoma cohorts, finding a strong positive correlation between expression levels of the *AR* and *EGFR* genes in multiple data sets (Fig. 7c). In the TCGA repository, we stratified lesions

according to expression scores of the *AR, EGFR, SERPINE1*, and *AXL* genes (Fig. 7d). *AR* expression was positively associated with *EGFR* in both primary and metastatic melanoma lesions from male as well as female patients (Fig. 7e). *AR* expression also positively correlated with *SERPINE1* and *AXL* levels in the metastatic but not primary lesions, in keeping with the complex role played by these genes in melanoma progression (Fig. 7e). Lesions were subdivided according to optimal cutoff levels of average *AR* and *EGFR* expression, with patients with tumors with higher average expression levels having significantly lower survival than those with negative ones (log-rank test, *p* = 0.0026) (Fig. 7f). The findings remained significant after correcting for age, sex, and primary or metastatic status (multivariate Cox regression, *p* = 0.0073).

Thus, increased AR expression in melanoma cells elicits changes found in a BRAFi-tolerant subpopulation and enhanced *EGFR* and *SERPINE1* expression of likely clinical significance.

### Targeting AR overcomes BRAFi resistance

The above results suggested that AR is a positive determinant of melanoma progression and BRAFi resistance.

To assess whether targeting of AR in BRAFi-resistant melanoma cells elicits opposite effects, we treated primary and established melanoma cells with two different AR inhibitors; AZD3514, suppressing AR activity through both ligand competitive and non-competitive mechanisms[34], and ARCC4 causing PROTAC-mediated degradation[26]. RT-qPCR and IF analysis showed that treatment with both inhibitors caused effective loss of AR expression (Fig. 8a, Supplementary Fig. 7a, b), which was paralleled by a decrease of SERPINE1 as well as EGFR expression (Fig. 8b, Supplementary Fig. 7c). Similar results were observed when silencing *AR* in three BRAFi-resistant melanoma lines by two different shRNA vectors (Supplementary Fig. 7d).

The findings are of functional significance, as live-cell imaging assays showed that treatment with either AR inhibitors blunted the proliferation of BRAFi-resistant melanoma cells and, at the same time, induced cell death (Fig. 8c, d). In contrast to the inhibitory effects of AR inhibitors on the growth of BRAFi-resistant melanoma cells, the treatment with the AR agonist dihydrotestosterone (DHT) significantly increased their proliferation (Supplementary Fig. 7e).

To assess whether inhibition of AR activity could also prevent the emergence of BRAFi resistance, drug-naive melanoma cells were cultured in the presence of DAB alone or in combination with the AR inhibitors. Consistent with previous studies[16,35], a large number of BRAFi-resistant colonies emerged in cultures of parental melanoma cells treated with the BRAFi alone, which was significantly reduced in cultures concomitantly treated with AR inhibitors (Fig. 8e).

The studies were extended to an orthotopic model of melanoma development based on intradermal Matrigel injection of cells into immunodeficient mice. BRAFi-resistant A375 cells were treated with AZD3514 (10 μM) or DMSO vehicle 24 h before injection. As shown in Fig. 8f, g, this single exposure to the AR inhibitor was sufficient to drastically reduce the tumorigenicity of cells, which formed lesions with significantly reduced cell density and proliferation relative to controls.

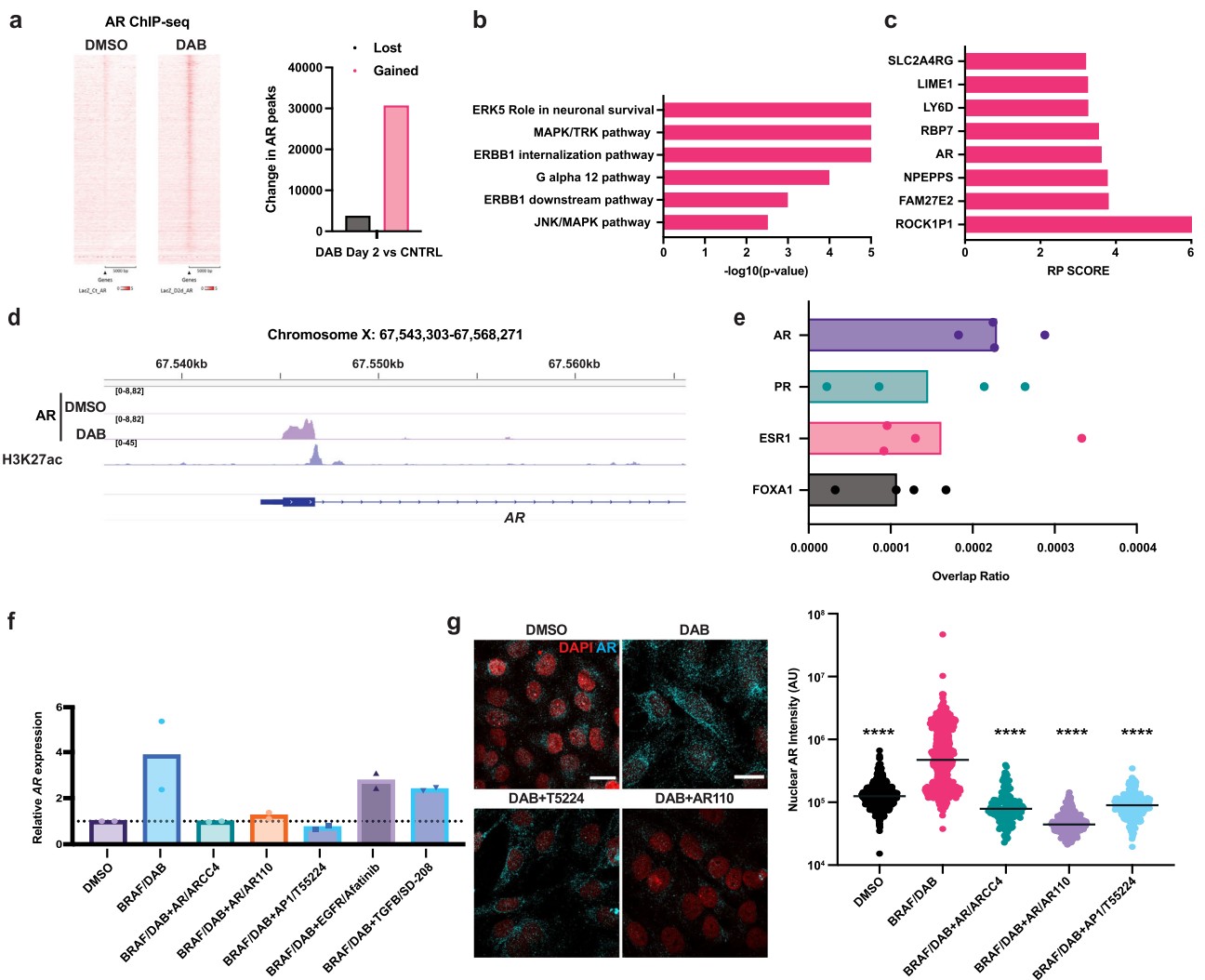

**Fig. 3 | BRAFi treatment of melanoma cells increases AR expression in melanoma cells through an AR-positive feedback loop. a** Tornado plots of AR binding sites, as determined by ChIP-seq analysis, 5,000 bp upstream and downstream of TSS, in A375 cells treated for 48 hrs with DAB versus vehicle (DMSO). The heatmap depicts the ChIP-seq signal at lower (white) and higher (red) peak intensity. Quantification of lost and gained AR binding peaks in DAB-treated versus control cells. **b** Functionally relevant gene ontology (GO) families of the genes significantly upregulated by BRAFi treatment in three melanoma lines and bound by AR (by ChIP seq) in DAB-treated A375 cells. The p-value ($-\log_{10}$) is indicated on the x-axis, unpaired two-tailed t-test. The differentially expressed genes are provided in Supplementary Data 2. **c** Top target genes bound by AR in DAB-treated A375 cells, as identified by ChIP-seq, using the Cistrome DB toolkit (http://dbtoolkit.cistrome.org/). RP = Regulatory potential. **d** Illustration of AR-binding peaks within AR locus, determined by ChIP-seq and displayed using the integrative genomic viewer software (IGV), in DAB-treated A375 cells (deep purple) versus control (cyan). H3K27ac peaks (light purple), map histone modifications overlapping with AR binding regions, were derived from[54]. **e** Top transcription factors with predicted overlapping binding with AR to the AR locus using the Cistrome DB toolkit (http://dbtoolkit.cistrome.org/). Each dot represents the average overlap ratio derived from individual studies. y-axis: names of transcription factors; x-axis: overlap ratio values. **f** RT-qPCR analysis of AR expression in A375 melanoma cells treated with inhibitors of the indicated molecules/pathways for 48 h. AR expression (arbitrary units) relative to untreated controls, after *RPLP0* normalization. Mean ± SD, *n* (biological replicates)=2. **g** AR IF in A375 melanoma cells treated with inhibitors of the indicated molecules/pathways (48 h). Representative images and quantification of AR nuclear signal (arbitrary units). Mean ± SD, *n* (cells/sample)>200, unpaired, two-tailed *t*-test, ****$p < 0.0001$. Color scale: red, DAPI; cyan, AR. Scale bar: 20 μm.

To extend the work to a syngeneic mouse model, we started by selecting BRAFi-resistant mouse melanoma cells (YUMM1.7BR)[36] by the same multistep protocol of culturing at increasing concentrations of DAB as for the human cells. RT-qPCR and immunoblot analysis showed that, even for the mouse cells, the acquisition of BRAFi resistance was accompanied by the concomitant up-regulation of the *AR*, *EGFR*, and *SERPINE1* (PAI-1) (Fig. 9a, b). Expression of all these genes was suppressed by treatment of the BRAFi-resistant cells with the AR inhibitors (Fig. 9c) which, at the same time, suppressed their proliferation (Fig. 9d).

In a first set of tumorigenicity assays, YUMM1.7BR cells were pretreated with AZD3514 or ARCC4 in parallel with DMSO control for 12 hours prior to injection into immunocompetent mice. Melanoma

cells pretreated with the AR inhibitors produced tumors of significantly smaller size than controls as observed with the human cells (Fig. 9e). The work was extended by assessing the impact of systemic administration of AR inhibitors on tumorigenesis. Mice injected with the YUMM1.7BR cells were treated with AZD3514, ARCC4 and Enzalutamide, another AR inhibitor[37], by gavage, starting at day 3 after injection. Tumor size expansion was significantly reduced at later times in mice treated with all three AR inhibitors (Fig. 9f). RNA extraction and RT-qPCR analysis of lesions at the end of the experiment showed that even in vivo AR inhibition resulted in the reduced expression of the *EGFR* and *SERPINE1* genes (Fig. 9g). Parallel IF analysis showed a striking decrease in Ki67 labeling index and caspase 3 positivity of tumors in mice treated with the AR inhibitors versus controls,

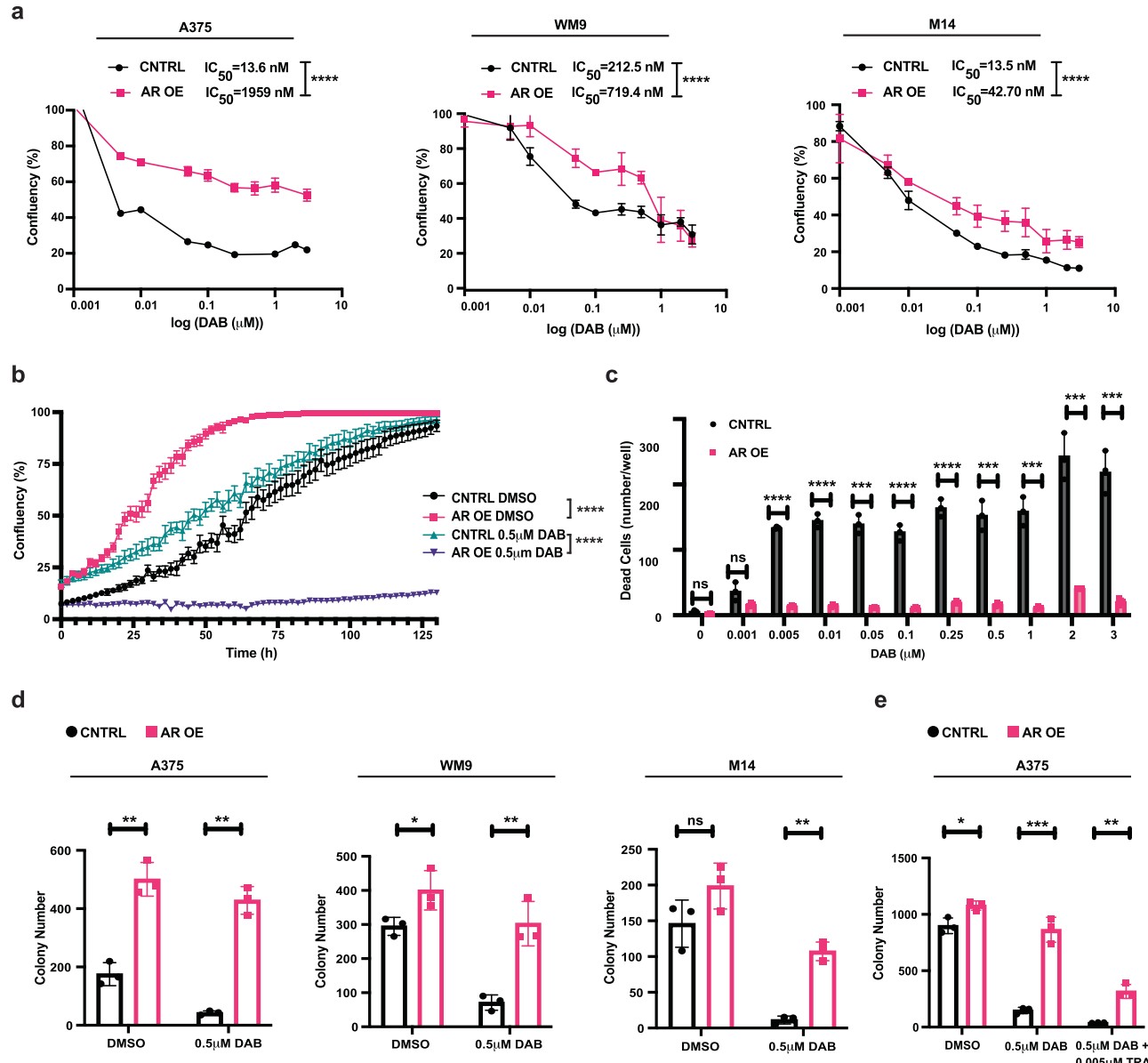

**Fig. 4 | AR overexpression confers BRAFi resistance. a** Cell density assays (CellTiter-Glo) of the indicated melanoma cells stably infected with an AR over-expressing lentivirus (AR OE) versus LacZ expressing control (CNTRL) and treated with the indicated increasing concentrations of DAB for 72 h. The calculated $IC_{50}$ for each condition is indicated above. Mean ± SD, n(biological replicates/condition) =3, Log-rank test, ****$p < 0.0001$. **b** Live-cell imaging proliferation assays (IncuCyte) of AR overexpressing (AR OE) versus control (CNTRL) A375 melanoma cells obtained as in **a** cultured with the indicated concentrations of DAB or DMSO. n(biological replicates) = 3, (four images per replicate every 4 h for 128 h). mean ± SD. Pearson r correlation test. ****$p < 0.0001$. **c** Live detection of BRAFi-induced cell death (IncuCyte, Cytotox Red) of AR overexpressing (AR OE) versus control (CNTRL) melanoma cells (A375) at 72 h of treatment with DAB. n(biological replicates) = 3, (four images per replicate every 4 h for 128 h). Mean ± SD, unpaired, two-tailed $t$-test, ns, non-significant, ***$p < 0.001$; ****$p < 0.0001$. **d** Clonogenicity

assays of the indicated melanoma cells transduced with an AR overexpressing lentivirus (AR OE) versus empty vector control (CNTRL) and treated with DAB (0.5 µM) or DMSO. Cells were plated in triplicates at low density (5000 cells/6 cm dish) and quantified after one week following crystal violet staining. n(dishes/condition)=3, unpaired, two-tailed $t$-test, Mean ± SD. A375: (DMSO AR OE vs DMSO CNTRL) $p = 0.00130$ and (DAB AR OE vs DAB CNTRL) $p = 0.000152$; M14: (DMSO AR OE vs DMSO CNTRL) p = 0.1182 and (DAB AR OE vs DAB CNTRL) p = 0.00029; WM9 (DMSO AR OE vs DMSO CNTRL) $p = 0.0091$ and (DAB AR OE vs DAB CNTRL) $p = 0.0043$, ns: non-significant. **e** Clonogenicity assays of AR-overexpressing versus control A375 melanoma cells as in the previous panel treated with DAB (0.5 µM) individually and in combination with the MEK inhibitor Trametinib (5 nM). n(dishes/condition)=3, unpaired, two-tailed $t$-test, Mean ± SD. (DMSO AR OE vs DMSO CNTRL) $p = 0.018$; (DAB AR OE vs DAB CNTRL) $p = 0.00041$; (DAB + TRA AR OE vs DAB + TRA CNTRL) $p = 0.0015$.

consistent with the intrinsic suppression of proliferation and increased cell death observed in culture upon AR inhibition (Fig. 9h).

Intra-tumoral localization of immune cells is a key determinant of the protective anti-tumor response[38]. FACS analysis of dissociated cells from tumors at the end of the experiment revealed no significant changes in total numbers of CD8+ and CD4 + T cells expressing various markers of functional activity (CD44, FOXP3) and exhaustion (PD-1, Tim-

3, LAG-3). Rather, combined IF and 3D image reconstruction analysis showed a consistent and marked increase of clusters of CD8+ T cells, with interspersed CD4+ in multiple tumors of mice with systemic drug administration of AR inhibitors (Fig. 9i, Supplementary Fig. 8a).

The clusters of CD8+ cells were strongly positive for granzyme B expression, a family of serine proteases and marker of cytotoxic T lymphocytes that mediate their killing effects[39], with concomitant

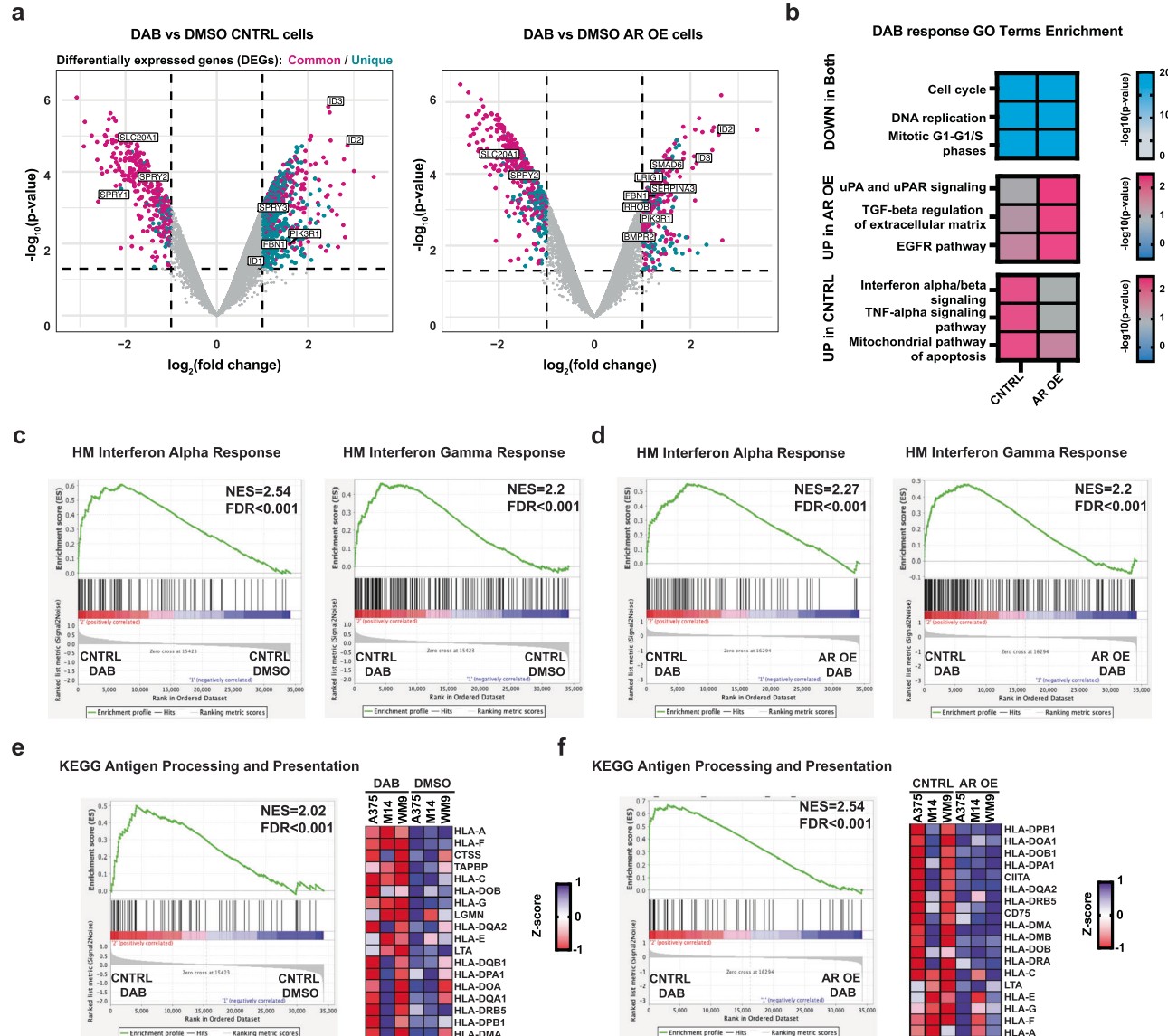

**Fig. 5 | Increased AR expression perturbs the transcriptional response of melanoma cells to BRAFi. a** Transcriptional response of melanoma cells plus/minus AR overexpression to acute BRAFi treatment. Volcano plot of transcriptional changes consistently elicited in A375, M14, and WM9 melanoma cells infected with control (LacZ expressing) (left) or AR overexpressing (right) lentiviruses by 48 h of treatment with Dabrafenib (0.5 μM) versus DMSO. The x-axis shows the fold change (log₂), and the y-axis shows the p-value (−log₁₀). Colored dots correspond to genes with a p-value < 0.05, two-tailed t-test, and log₂fold-change threshold of −1 and 1. Magenta and cyan dots correspond to genes similarly and specifically modulated by DAB treatment in control versus AR overexpressing melanoma cells, respectively. Indicated are TGF-ß and EGFR families. The list of modulated genes in the three melanoma cell lines is provided in Supplementary Data 2. **b** Functionally relevant GO families significantly downmodulated by BRAFi treatment in both control and AR overexpressing cells (upper), and GO families upmodulated only in

AR overexpressing cells (middle) or in control cells (bottom). The -log₁₀(p-value) is indicated by the heatmap color scale, two-tailed t-test. The list of modulated gene families is in Supplementary Data 2. **c–f** GSEA of transcriptional profiles of control melanoma cells (A375, M14, and WM9) plus/minus DAB treatment and of DAB-treated control versus AR overexpressing cells using predefined gene signatures of interferon alpha and gamma response **c**, **d** and antigen processing and presentation **e**, **f** derived from the hallmark gene set (HM)[22] and KEGG[55] collections. Genes are ranked by signal-to-noise ratio in DAB versus DMSO-treated melanoma cells; the position of individual genes is indicated by black vertical bars; the enrichment pattern is in green. In **e**, **f**, GSEA and the leading-edge analysis of the antigen processing and presentation signature are shown in each of the three melanoma lines plus/minus DAB treatment **e** and of DAB-treated control versus AR over-expressing cells **f**.

localized staining of caspase 3 positive tumor cells (Fig. 9k, Supplementary Fig. 8b). The results were quantified and extended to tumors formed by YUMM1.7BR cells pretreated with AR inhibitors, showing even in this case, increased clustering of CD8 + T cells (Fig. 9l, m, Supplementary Fig. 8d, e).

Hence, pharmacological inhibition of AR activity in BRAFi-resistant cells provides a tool to effectively suppress *EGFR* and *SERPINE1* expression, proliferation, and tumorigenicity with a concomitant enhancement of clustered T-cell infiltration and killing.

## Discussion

Resilience to cancer therapy remains a major challenge even with improved approaches[40]. In concert with modulation of the micro-environment, the resistance of cancer cells to targeted therapies can result from two mechanisms: (i) an intrinsic adaptive response, with the expansion of pre-existing cell populations; (ii) acquired resistance, through de novo genetic/epigenetic events[40]. The adaptive response, which can be very rapid, is the result of compensatory feedback mechanisms of therapeutic interest[41]. Drivers of adaptive responses

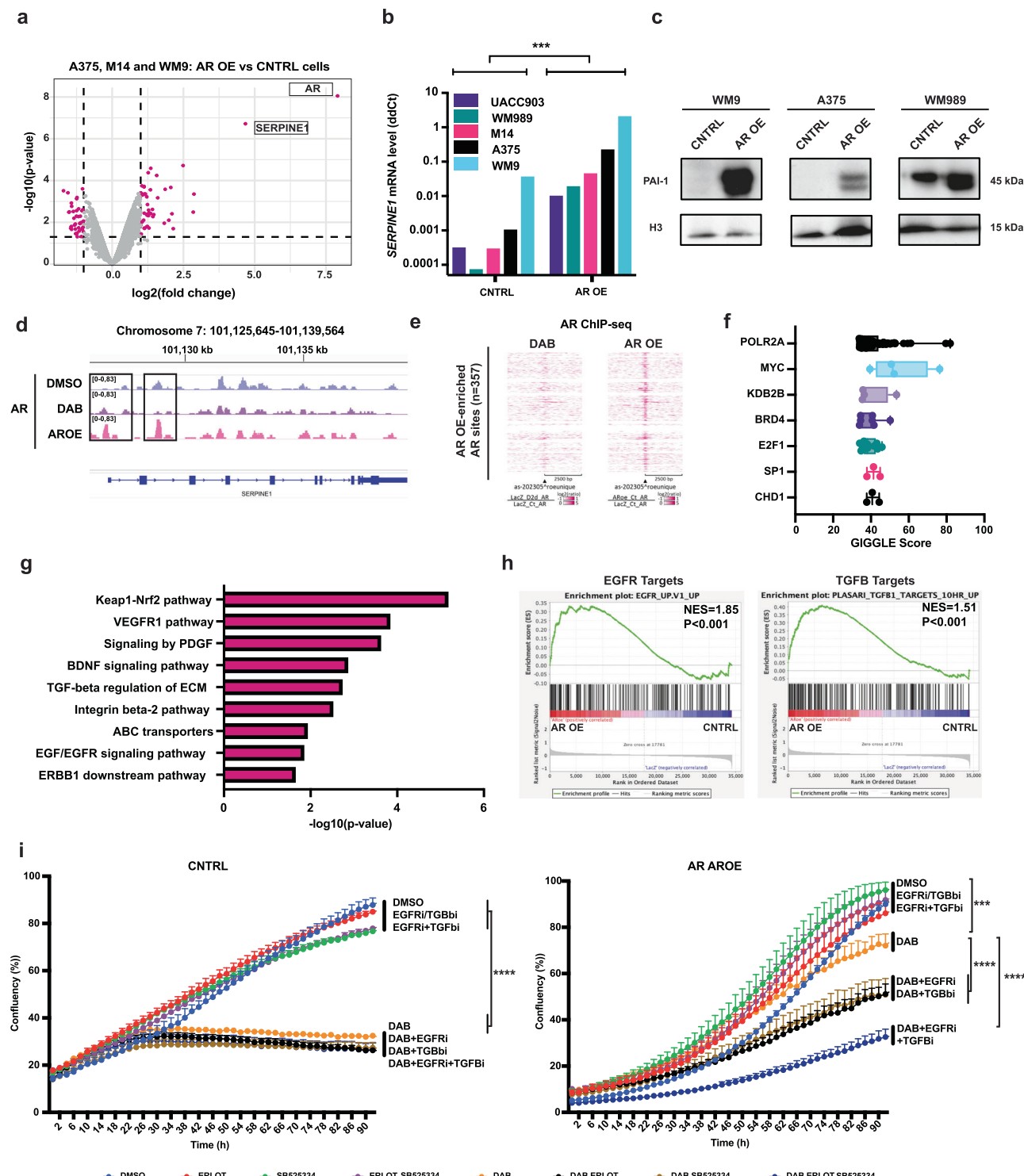

are typically involved in regulatory circuits of both normal and cancer cells and can be most effectively targeted in adjuvant therapy[40]. Our combined findings with melanoma cells from both male and female patients indicate that the AR is one such driver as a key determinant of the adaptive response to targeted therapy in both sexes, which may be used to prevent or delay resistance.

The initial sensitivity of melanomas with activating BRAF mutations to BRAF inhibitors can be overcome by several mechanisms, including the compensatory upregulation of the EGFR tyrosine kinase coupled with downmodulation of the MITF and SOX10 transcription factors[16,20,24,41]. In contrast to the negative role played by these

transcription factors, we have found that AR is a positive determinant of BRAFi resistance and EGFR expression. We previously showed that basal AR activity is required for sustained proliferation and tumorigenesis of melanoma cells, with AR functioning as a bridge between RNA-Pol II and DNA repair proteins and ensuring the continuous DNA repair process associated with gene transcription[9]. The markedly increased AR expression that is already occurring at early times of BRAFi and MEKi exposure suggested that this molecule can fulfill a second distinct function in melanoma cells as part of an adaptive mechanism leading to targeted drug resistance. In fact, a comparative analysis of transcriptomic profiles revealed that persistently increased

**Fig. 6 | Increased TGF-ß and EGFR signaling can account for DAB resistance resulting from increased AR expression. a** Transcriptional changes elicited in melanoma cells by AR overexpression. Volcano plots of genes similarly modulated in A375, M14, and WM9 melanoma cells stably infected with AR- versus LacZ- (control) expressing lentiviruses under control conditions (without DAB treatment). Colored dots (magenta) correspond to genes with fold change (log₂) thresholds of −1 and 1 and p-value < 0.05, two-tailed t-test. The list of differentially expressed genes is in Supplementary Data 3. **b** RT-qPCR analysis of *SERPINE1* expression, normalized to *RPLPO*, in melanoma cell lines stably infected with an AR overexpressing lentivirus versus LacZ control (the same cells as in Fig. 4). *n*(biological replicate)=5, paired, two-tailed t-test, *p = 0.0003. **c** PAI-1 and Histone H3 WB analysis of melanoma cell lines stably infected with an AR overexpressing lentivirus or LacZ control. **d** Illustration of AR binding peaks in A375 cells onto the *SERPINE1* gene and promoter region (squares) as determined by ChIP-seq, with IGV software. AR-binding peaks in DMSO control (blue), DAB-treated (purple), and AR OE (magenta) A375 cells. **e** Tornado plots visualizing AR binding sites, as determined by ChIP-seq, focusing on the TSS (±2500 bp), of genes upregulated in AR

overexpressing (right) or DAB-treated (left) A375 melanoma cells. The heatmap depicts the ChIP-seq signal normalized to the untreated, with levels of red indicating peak intensity. **f** Binding sites of transcription factors and chromatin regulators derived from public ChIP-seq profiles and the Cistrome toolkit (http://dbtoolkit.cistrome.org/) that overlap with AR-bound sites identified by ChIP-seq of AR-overexpressing cells. The GIGGLE score is indicated on the x-axis, n(SP1 and CHD1) = 3, n(MYC and KDB2B) = 4, *n*(BRD4) = 7, *n*(E2F1) = 13, *n*(POLR2A) = 98. Box plots show the median, with the edges delineating 25th and 75th percentiles. **g** GO analysis of genes bound by AR and upregulated in AR-overexpressing cells. The *p*-value (-log₁₀) is indicated on the x-axis, two-tailed t-test. **h** GSEA of transcriptional profiles of cell lines infected with an AR overexpressing versus LacZ control lentivirus using TGF-ß and EGFR gene signatures. Black vertical bars indicate individual genes, the enrichment pattern is in green, permutation-based *p* values. **i** Live-cell imaging proliferation analysis (IncuCyte) of AR overexpressing (AR OE) versus control (CNTRL) A375 melanoma cells, treated with Dabrafenib or DMSO and/or inhibitors of the EGFR, TGF-ß and BRAF. Mean ± SD. *n*(cultures/condition)=3; Pearson r correlation test, **p = 0.003.

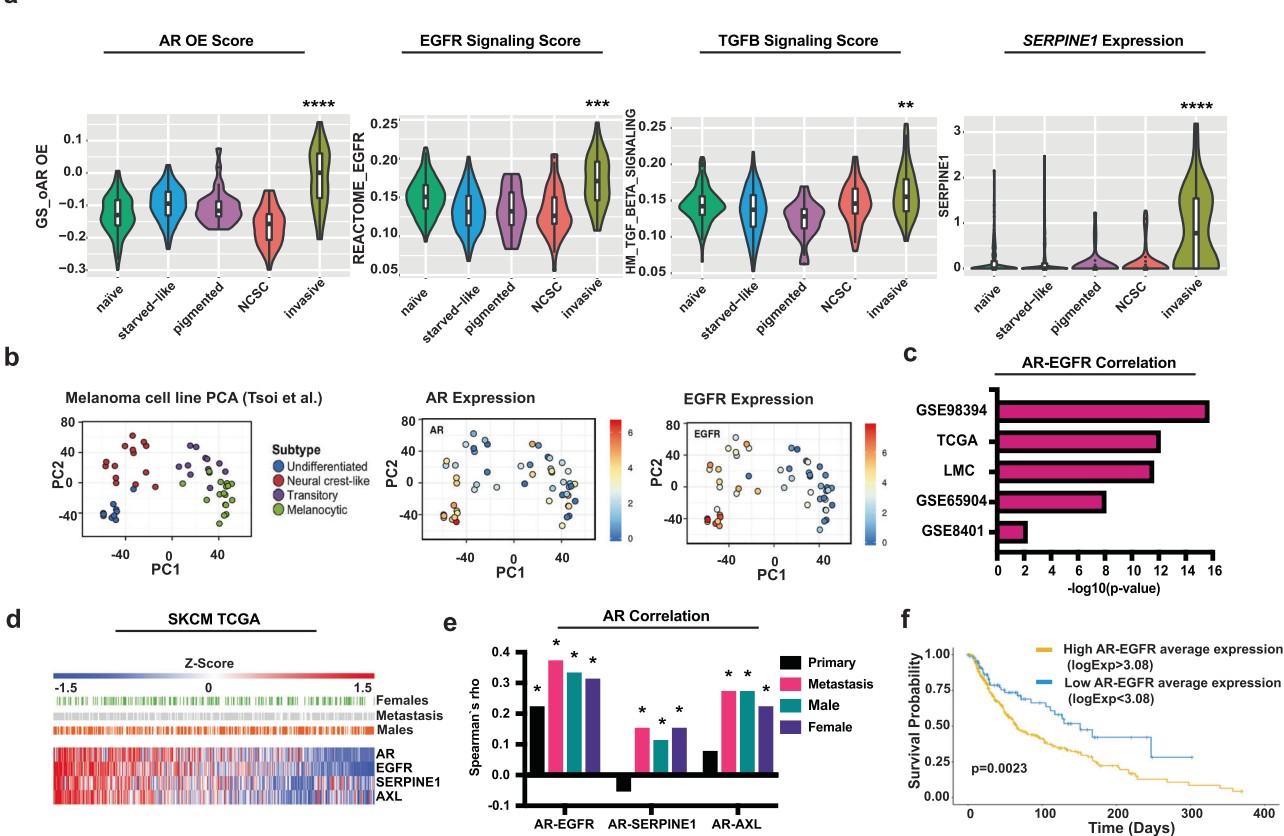

**Fig. 7 | Increased AR expression elicits transcriptional changes of clinical significance found in BRAFi-resistant subpopulations. a** Scores of AR, EGFR, TGF-ß gene signatures, and *SERPINE1* expression in cell subpopulations identified by single-cell RNA-seq analysis of a PDX model of melanoma BRAFi-resistance[21]. We established a gene signature resulting from AR overexpression in melanoma cells (19 upregulated and 39 downregulated genes, p-value < 0.01, absolute FC >1, Supplementary Data 3) and calculated the scores using AUCell[51], in reported single cell profiles of drug-naive melanoma cells and BRAFi-induced starved-like (SMC), pigmented, invasive and neural crest-like subpopulations[21]. We performed similar score calculations with the Reactome signaling pathway (EGFR)[56] and hallmark EGFR and TGF-ß gene set signatures[22], and single gene *SERPINE1* expression levels. Violin plots show individual cell score median, box (25–75%) and whiskers (5–95%). n(single cells)=486, the significance between invasive versus naive cell populations (mean) was calculated by Welch's *t*-test[52]. **p < 0.01; ***p < 0.001; ****p < 0.0001. **b** *AR* and *EGFR* expression in melanoma cell lines previously clustered according to differentiation trajectories[23]. Principal Component Analysis (PCA) of the expression profiles of individual melanoma cell lines (dots) and corresponding subtypes.

Overlapping color-coded indicate *AR* and *EGFR* levels derived from (http://systems.crump.ucla.edu/dediff/). **c** Correlation between *AR* and *EGFR* expression from melanoma clinical cohorts: GSE98394 (n = 51 primary melanomas); TCGA (*n* = 472 primary melanomas and metastases); LMC (*n* = 703 primary melanomas); GSE65904 (*n* = 214 melanoma metastases); GSE8401 (*n* = 83 primary melanomas and metastases from xenograft models). p-value (−log₁₀), using *corrplot* v0.92 package with Spearman's correlation. **d** Heatmap of indicated genes expression (Z-scores) in individual melanoma from TCGA Firehose Legacy, (February 2022). n(melanoma)=472. Z-scores from median-centering expression values (log₂) and divided by standard deviation. For each lesion is shown patients' sex, and if from metastasis. **e** Correlation between *AR* an*d EGFR*, *SERPINE1* and *AXL* expression from the cohort of **d**, using the *corrplot* 0.92 package and Spearman's correlation. Rho coefficients, *=*p*-value < 0.05. **f** Kaplan-Meier curves of melanoma patients' long-term survival from the TCGA dataset, divided by high (yellow line) versus low (blue line) levels of *AR* and *EGFR* expression, calculated using the optimal cutpoint for continuous variables (log₂ (Expression value) = 3.08), obtained from the maximally selected rank statistics from the *maxstat* R package.

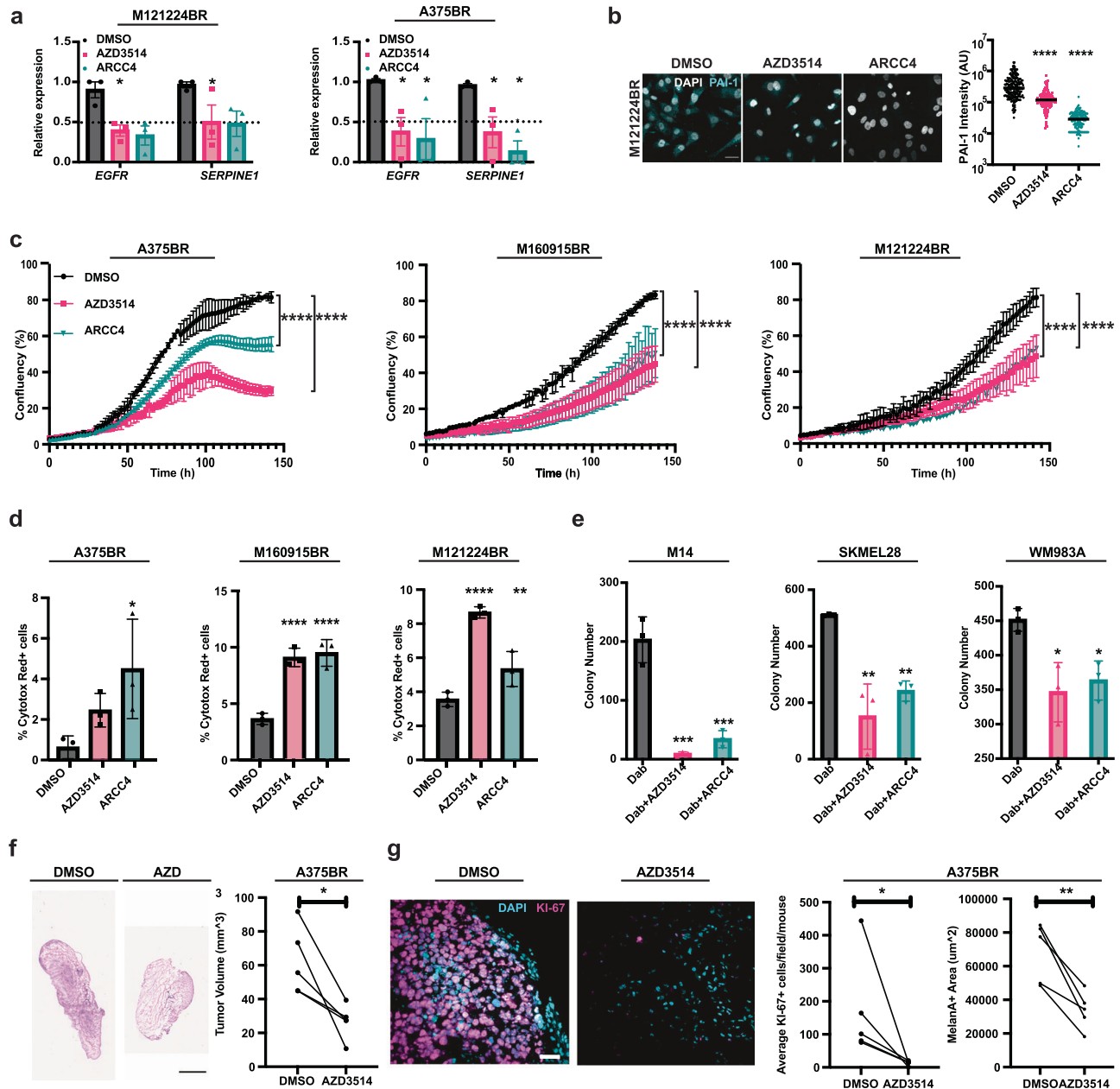

**Fig. 8 | Targeting AR overcomes BRAFi resistance. a** RT-qPCR analysis of *EGFR* and *SERPINE1* expression normalized to *RPLPO*, in two melanoma cells (M121224BR and WM983ABR) propagated in the presence of Dabrafenib and treated with the AR inhibitors AZD3514 and ARCC4 or DMSO for 48 h. Mean ± SD n(biological replicates)=3, one-way ANOVA, M121224BR *p* = 0.0114; WM983ABR *p* = 0.0123. **b** IF with anti-PAI-1 antibodies of melanoma cells (M121224BR) treated with the AR inhibitors AZD3514 or ARCC4 or DMSO control for 48 h as in the previous panel. Representative images and quantification of the PAI-1signal per individual cells (arbitrary units). n(cells per sample)≥100, unpaired, two-tailed *t*-test, **** *p* < 0.0001. Color scale: gray, DAPI; cyan, PAI-1. Scale bar: 40 µm. AR IF in parallel cultures is in Supplementary Fig. 6. **c** Live-cell imaging proliferation assays (Incu-Cyte) of the indicated BRAFi-resistant cells treated with AR inhibitors or DMSO, as in **a**, **b**. n(biological replicates) = 3, (four images per well every 4 h for 150 h). Mean ± SD; Pearson r correlation test, ****p* < 0.0001. **d** Cell death as detected by live-cell staining (IncuCyte, Cytotox Red) of cells as in the previous panel at 72 h of

treatment with the AR inhibitors versus DMSO control. Four images per cell culture, *n* (biological replicates)=3, median, box (25-75%) and whiskers (5-95%), unpaired *t*-test, *, *p* < 0.05; **p* < 0.01; ****p* < 0.0001. **e** Clonogenicity assays of three drug-naive melanoma cell lines treated with Dabrafenib (0.5 µM) individually or in combination with AZD3514 (10 µM) or ARCC4 (1 µM). Quantification of crystal violet stained colonies. n(dishes)=3, Mean ± SD, unpaired, two-tailed *t*-test, (M14) ****p* = 0.0001; (SKMEL28) **p* = 0.0018; WM983A) **p* = 0.0130. **f** Tumor volume quantification 14 days after intradermal injection into immunodeficient mice (NOD.CB17-Prkdcscid/ J) of BRAFi-resistant A375 cells pretreated with AZD3514 (10 µM) or DMSO vehicle 24 h prior to injection. n(tumors)=5, Paired, two-tailed t-test, **p* < 0.05. Scale bar: 1 mm. **g** IF of tumors as in **f** using anti-Ki-67- and anti-MelanA antibodies. Representative images and quantification of Ki67+ cells and MelanA+ area in arbitrary units. n(tumors)=5, Paired, two-tailed t-test, **p* < 0.05, **p* = 0.0063. Color scale: cyan, DAPI; magenta, Ki-67. Scale bar: 40 µm.

AR expression in melanoma cells modulates different sets of genes from those affected by *AR* gene silencing[9] (see Supplementary Fig. 9 for a comparison).

Regarding the mechanisms involved in the induction of AR expression, immunofluorescence analysis demonstrated that this

occurs in the vast majority of melanoma cells at early times to DAB treatment. Additionally, an increase in nuclear AR protein levels was observed, which was further confirmed through cell fractionation studies. Paralleling these results, ChIP-seq analysis revealed a significant augmentation in AR binding to target genes induced in

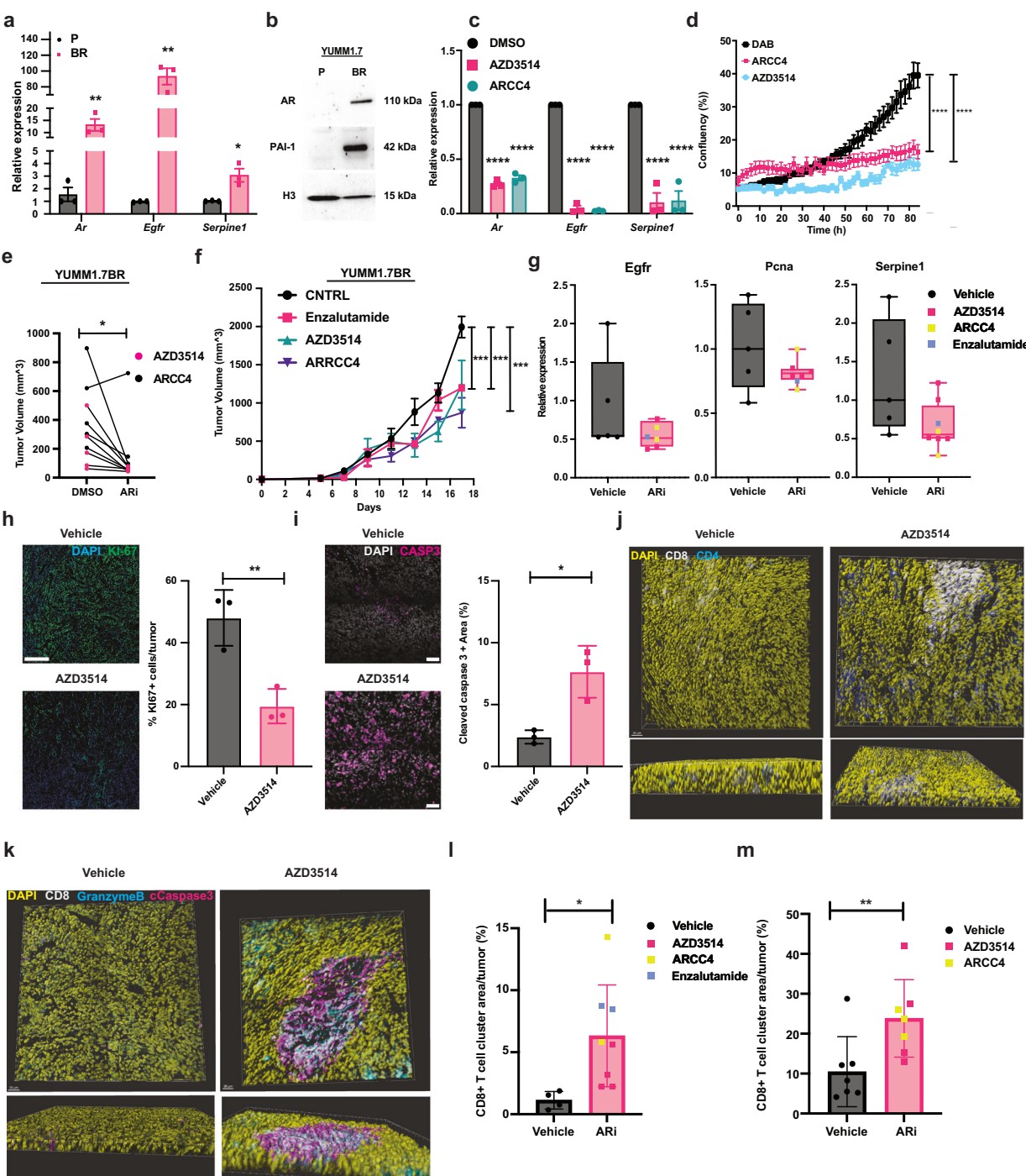

melanoma cells after 48 h of DAB treatment, including the *AR* gene itself. The induction of *AR* mRNA expression by DAB treatment was entirely suppressed by AR inhibitors, suggesting the involvement of a positive feedback loop mechanism. Further investigation will be required to explore possible underlying mechanisms, including post-transcriptional modifications of the AR protein and/or its dissociation from cytoplasmic retaining proteins.

While short-term exposure of melanoma cells to BRAF inhibitors induces a notable increase in AR expression, this induction alone is insufficient to cause resistance to BRAF inhibitors, as it resulted instead from sustained AR expression. Importantly, persistently elevated AR expression in melanoma cells did not block but rather subverted the transcriptional response of melanoma cells to BRAF inhibition. Underlying the different sensitivity, apoptosis-related genes were

induced by BRAFi treatment to a much greater extent in control than AR overexpressing cells. The efficacy of BRAF inhibitors depends on triggering a cancer cell death program associated with an impact on the tumor immune microenvironment[42]. Gene signatures related to interferon signaling, inflammation, and antigen presentation, which can enhance immune stimulation and response to checkpoint inhibitors[43], were all induced by BRAFi treatment of control but not AR-overexpressing cells. A cross-connection has been established between BRAFi resistance and poor response to immune checkpoint control that does not depend on selection by the immune system and is a cancer cell-instructed[13], in which increased AR expression may be involved. Consistent with this possibility, in a syngeneic mouse model with BRAFi-resistant melanoma cells, which exhibited a similar modulation of AR expression and activity as the human cells, suppression

**Fig. 9 | AR inhibition suppresses tumorigenicity of BRAFi-resistant melanoma cells. a** RT-qPCR of the indicated genes, normalized to *RPLPO*, in YUMM1.7 mouse melanoma cells, parental (P), or selected for DAB resistance (BR). Mean ± SD, n(biological replicates)=3, unpaired, two-tailed *t*-test, (AR)**p = 0.0098; (EGFR) **p = 0.00086; (SERPINE) *p = 0.017. **b** WB of AR, PAI-1and H3 in (P) versus (BR) YUMM1.7 cells. **c** RT-qPCR of the indicated genes in YUMM1.7-BR cells treated with AZD3514 or ARCC4 AR inhibitors (ARi), or DMSO. Mean ± SEM, n(biological replicates)=3, one-way ANOVA, ****p < 0.0001. **d** Live-cell imaging proliferation assays (IncuCyte) of YUMM1.7-BR cells treated with DAB versus AR inhibitors. n(dishes)= 3, mean ± SD, Pearson r correlation test, ****p < 0.0001. **e** Tumor volume quantification 14 days after intradermal injection into immunocompetent mice (C57BL/6 J) of YUMM1.7-BR cells, pretreated (12 hours) with either AZD3514 or ARCC4 AR inhibitors (ARi) or DMSO. n(tumors) = 5, paired, two-tailed *t*-test, **p = 0.0014. **f** Tumor size (volume) in immunocompetent mice (C57BL/6 J) injected with YUMM1.7-BR cells followed by *gavage* with the indicated AR inhibitors or DMSO control (CNTRL), starting day 3 after injection. n(tumors)= 5, mean ± SD, two-tailed Pearson r correlation test, ***p = 0.0001. **g** RT-qPCR expression analysis of the indicated genes, normalized to *RplpO*, in tumors as in **f**. Median, box (25–75%) and whiskers (5-95%), n(tumors) = 5. **h** Ki67 IF (green) of tumors as in **f**, with DAPI (blue). Representative images and quantification of Ki67[+] cells per tumor. Mean ± SEM, n(tumors)= 3, unpaired, two-tailed *t*-test, **p = 0.0096. Scale bar: 250 μm. **i** Cleaved-caspase3 IF (magenta) of tumors as in **f**, with DAPI (blue). Representative images and quantification of cleaved-caspase3[+] areas per tumor. Mean ± SEM, n(tumors)= 3, unpaired, two-tailed *t*-test, **p < 0.01. Scale bar: 50 μm. **j** CD4+ (blue) and CD8+ (white) IF of sections from tumors made as in **f** and 3D reconstruction analysis by Imaris software. Additional tumors are in Supplementary Fig. 8a. Scale bar: 30 μm. **k** Cleaved-caspase3 (magenta), granzyme B (cyan), CD8 (gray) IF of tumors made as in **f**, with DAPI (yellow), followed by a 3D analysis. Additional tumors are in Supplementary Fig. 8b. Scale bar: 30 μm. **l, m** Quantification of the tumor areas occupied by CD8[+] cell clusters (*n* (cells/cluster) >5), as detected by IF of tumors formed in mice with *gavage* **l** or by ARi-treated cells **m**, as those from **f**, **e**, respectively. Mean ± SEM, unpaired, two-tailed, t-test in **l**, Veh n(tumors)=4; ARi n(tumors)=8; *p = 0.0188, and paired, two-tailed t-test in **m** Veh n(tumors)=8; ARi n(tumors)=8, ***p < 0.0009. Images used for quantifications are in Supplementary Fig. 9c–e.

of tumorigenicity by AR inhibitors was accompanied by increased infiltration and clusters of granzyme-positive CD8 T cells with adjoined areas of apoptotic (caspase 3-positive) tumor cells.

Besides suppressing the induction of pro-apoptotic and immunomodulatory genes, elevated AR expression led to the upregulation of gene families associated with resistance to BRAF inhibitor (BRAFi) treatment, specifically EGFR and TGF-ß related genes. Under basal conditions, AR overexpression also induced gene signatures of two pathways, with *SERPINE1*, an established TGF-ß target with pro-tumorigenic functions[30,31], being the most prominently upregulated gene. Notably, EGFR- and TGF-ß-related genes, including SERPINE1, were selectively bound and upregulated by AR in cells with sustained expression of the AR gene, but not in cells with short-term upregulation (induced by DAB treatment). The selective targeting of these genes may be attributed to a complex cascade of events triggered by AR upregulation, which influences chromatin configuration and accessibility. Alternatively, it is possible that a pre-existing subpopulation with a specific chromatin configuration, exhibiting differential response to increased AR expression, becomes selectively amplified over time.

It has been recently shown that elevated *SERPINE1* expression in melanoma cells is associated with a bad prognosis and poor response to immune checkpoint inhibitors[30]. The positive connection between *AR* and *EGFR* and *SERPINE1* expression was validated by single-cell analysis of melanoma cells in a PDX model of BRAFi response: increased *AR* and *EGFR* gene signatures were coincidental with elevated *SERPINE1* expression in a specific BRAFi tolerant subpopulation characterized by high AXL expression and invasive features[21]. Similarly, in a study on the heterogeneity of melanoma cell lines and tumors, *AR* expression was found to cluster together with *EGFR* and *SERPINE1* in an undifferentiated AXL positive subgroup connected with targeted drug resistance[23]. The positive association of *AR* with *EGFR*, *SERPINE1*, and *AXL* expression was confirmed in a large patient cohort, irrespective of sex and primary versus metastatic lesions, with poor survival with tumors with elevated *AR-EGFR* levels.

AR has been intensely studied as a driver of metastatic prostate cancer, with resistance to AR-targeting approaches resulting from various mechanisms[44]. Sustained and/or deregulated AR activity is a point of convergence of multiple mechanisms[45], with comparative analysis of transcriptomic profiles revealing common and distinct sets of genes under AR control among various prostate cancer studies (GSEA MSigDB Database and[46]). AR-responsive gene signature derived mostly from prostate studies was highly enriched in transcriptomic profiles of various BRAFi-resistant melanoma lines (Fig. 1f). For a wider perspective, 17 different gene signatures measuring AR activity in prostate cells were used for GSEA of these profiles. Leading-edge analysis of the GSEA plots showed a significant number of genes that are commonly modulated in several prostate-derived gene signatures and in melanoma cells as a function of BRAFi resistance (Supplementary Fig. 9b and Supplementary Data 1).

Overall, genetic and epigenetic changes of *AR* resistance are less likely to occur in melanoma, in which other genes drive the disease. As also indicated by another study[15], inhibitors targeting AR activity and expression could therefore be employed as co-adjuvants to prevent/delay targeted drug resistance and, as we have shown, reduce tumorigenicity of BRAFi-resistant cells while at the same time enhancing CD8[+] T cells infiltration. Given the connection between BRAFi resistance and poor immune response[13], as well as the intrinsic role of AR activation in dampening the T cell activity[12,47], AR targeting may be beneficial in the treatment regimens with immune checkpoint inhibitors.

## Methods

### Study approvals

All human samples were obtained from surplus melanoma material collected from de-identified patients with written, informed consent to participate in the research (BASEC-Nr 2017−00494). Pre- and post-treatment metastatic melanoma sections were from the Live Cell Biobanks of the University Research Priority Program (URPP) "Translational Cancer Research" (Mitchell P. Levesque, University Hospital Zurich). We had no access to sensitive information.

All animal studies were carried out according to Swiss guidelines for the use of laboratory animals, with protocols approved by the University of Lausanne animal care and use committee and the veterinary office of Canton Vaud (animal license No. 1854.4 f/1854.5a).

### Cell culture

The list of melanoma cell lines and primary melanoma cells is provided in Supplementary Data S4. Early passage primary melanoma cell cultures (M160915 and M121224) were established from discarded melanoma tissue samples by the University Research Priority Program (URPP) Live Cell Biobank (University of Zurich) following institutional requirements. WM115, WM9, WM983A, WM989, UACC903, and UACC903BR melanoma cells were a gift from Dr. Meenhard Herlyn (The Wistar Institute, US). The YUMM1.7 melanoma cell line[36] was provided by Dr. Ping-Chih Ho (UNIL).

All melanoma cell lines and patient-derived primary melanoma cells were maintained in Roswell Park Memorial Institute (RPMI) medium (Thermo Fisher Scientific) supplemented with 10% (v/v) fetal bovine serum (Thermo Fisher Scientific). YUMM1.7 melanoma cells were maintained in Dulbecco's Modified Eagle Medium (DMEM) (Thermo Fisher Scientific) supplemented with 10% (v/v) fetal bovine serum (Thermo Fisher Scientific). All cell lines were routinely tested negative for *Mycoplasma*.

## Cell manipulations and treatments

Lentiviral particle productions and infections were performed, as described previously[9]. Melanoma cells were transduced with *AR* overexpressing (a gift of Dr. Karl-Henning Kalland, Bergen University, Bergen, Norway) or LacZ expressing control lentiviruses for 6 h. Two days post-infection cells were selected using 5 µg/ml of Blasticidin for 6 days. RNA or protein samples were collected 7 days after infection.

BRAFi-resistant (BR) cell lines were established from the parental (P) cells (A375, M160915, M121224, and WM983A) by culturing in Dabrafenib for a period of 4 weeks, with weekly multistep increases in concentration from 0.5 to 3 µM. Resistant cells were thereafter continuously cultured in the presence of 3 µM Dabrafenib.

For short-term in vitro experiments with various chemical inhibitors, 24 h post-seeding cells were treated with the following compounds at the indicated concentrations: Dabrafenib (0.5 µM), PLX-4720 (0.5 µM), Sorafenib (0.5 µM), Trametinib (0.005 µM), Cobimetinib (0.005 µM), s31-201 (50 µM), Bay 7085 (10 µM), T55224 (20 µM), sr11302 (10 µM), SU6668 (10 µM), SKI-606 (10 µM), CYT387 (10 µM), Erlotinib (5 µM), and SB52334 (5 µM). All inhibitors were purchased from SellectChem and dissolved in DMSO, which was used as vehicle control. RNA was collected 48 hours post-treatment.

For AR inhibition, 24 h post-seeding, melanoma cells were treated with 10 µM AZD3514 (SellectChem) or with 1 µM ARCC4 (Tocris). AR inhibitors were dissolved in DMSO, which was used as a control for all experiments.

For DHT treatment, 24 h post-seeding, melanoma cells were treated with 10 nM DHT (Sigma). DHT was dissolved in DMSO, which was used as a control for all experiments.

Cell proliferation/density assays were carried out by measuring ATP production using the CellTiter-Glo luminescent assay (Promega) as per the manufacturer's instructions. Dabrafenib dose-response curves and IC50 values were attained by fitting the curves to nonlinear regression with variable slope using GraphPad Prism.

For clonogenicity assays, cells were plated onto 60 mm dishes (10,000 cells/well; triplicate wells/condition) and treated the next day as indicated in the figure legends. Cells were cultured for 7 days for *AR* overexpressing experiments and 14 days for experiments with AR inhibitors. Colonies were fixed with methanol and stained with 1% crystal violet. The number of clones was counted using Fiji/ImageJ software.

For IncuCyte cell proliferation and cell death assays, 1000 melanoma cells per condition were seeded in triplicate in 96-wells plates. Drug treatments were applied 12 hours post-seeding, with cells allowed to proliferate for 5 days. Four independent images per well per condition were captured every 2 hours for 5 days. For cell death measurements, the IncuCyte Cytotox Red Reagent (Sartorius) was added to the cells according to the manufacturer's instructions. Cell confluence and Cytotox Red positive cells were quantified using the IncuCyte Zoom Live-Cell Imaging System (Sartorius).

For spheroid culture in 3D collagen lattices and sandwich assay, melanoma cells were cultured as spheroids in 3D collagen for 4-6 days, unless indicated otherwise. Multicellular spheroids were generated from subconfluent cells using the hanging-drop method. In brief, melanoma cells were resuspended in a complete RPMI medium supplemented with 15% or 20% methylcellulose (Sigma Aldrich) and incubated at a final concentration of 3000 cells/25 µl drop for 72 h. Next, the acellular layer of collagen matrix (1.8 mg/ml collagen type I solution (Corning)/10% FBS/1x EMEM/0.03M L-glutamine/0.015M NaHCO3) was first spotted on the surface multiwell plates and was allowed to pre polymerize (37 °C, 10 min). Meanwhile, melanoma spheroids were washed (PBS, 2 × 5 min) and embedded at the interphase of two layers of collagen matrix prior to collagen polymerization (37 °C, 1 h). Invasion type and efficacy were monitored by bright-field microscopy at 2, 4 days, and 6 days post-embedding. Image processing was performed using Fiji/ImageJ (http://fiji.sc/Fiji). The invasion was quantified as a 2D area of the invasive spheroid region from the bright field images at the abovementioned time points. The invasive area was obtained by subtracting the spheroid core area from the total spheroid area. Fusing spheroids and spheroids localized at the edge of the gel were excluded from the analysis.

## Immunofluorescence and immunohistochemistry staining

Immunofluorescence staining of tissue sections and cells was carried out as described previously[9]. In brief, paraffin-embedded sections were deparaffinized and rehydrated prior to a citrate-based buffer antigen retrieval. Frozen tissue sections (8 µm) or cultured cells were fixed in 4% paraformaldehyde (PFA) for 15 minutes at room temperature (RT). Samples were washed with PBS (3 × 5min) and permeabilized using 0.5% TritonX100 in PBS for 10 min. Samples were blocked using 2% bovine serum albumin in PBS for 1 h at RT. Primary antibodies were diluted in a blocking buffer (PBS/2% bovine serum albumin) and were incubated overnight at 4 °C. Following, samples were washed (PBS, 3 × 5min) and incubated with secondary donkey fluorescence conjugated secondary antibodies (Invitrogen) for 1 h at RT. DAPI was used to counterstain nuclei. Slides were washed (PBS, 3 × 5min) and mounted using Fluoromount Mounting Medium (Sigma-Aldrich). Control staining without the primary antibodies was performed in each case to subtract background and set image acquisition parameters.

Primary antibodies used were anti-androgen Receptor (#5153, Cell Signaling Technology, 1:200; #100272-MM05, (clone #5) Sino Biologicals, 1:200; #sc-377546, Santa Cruz Biotechnology, 1:200; #MA5-13426, ThermoFisher, 1:200), anti-EGFR (#AF23, Novus Biologicals, 1:200), anti-MelanA (#HPA048662, Sigma-Aldrich, 1:500 or Cat# sc-20032, Santa Cruz, 1:200), anti-PAI-1 (#19773; Novus Biologicals, 1:200), anti-Axl (#AF154, R&Dsystems, 1:500), anti-MHC class I (#sc-32235, Santa Cruz, 1:500), anti-CD8a (#550281, BD Biosciences, 1:1000), anti-granzyme B (#AF1865, R&Dsystems, 1:500), anti-cleaved caspase-3 (#9661, Cell Signaling Technology, 1:500), AF (Alexa Fluor) 647-conjugated anti-CD8 (#100727, BioLegend, 1:1000) and AF-488-conjugated anti-CD4 (#100425, BioLegend, 1:1000). Secondary antibodies used were AF 568-conjugated goat anti-mouse (#A-21202, ThermoFisher, 1:1000), AF 568-conjugated donkey anti-rabbit (#A-10042, ThermoFisher, 1:1000), AF 555-conjugated goat anti-rabbit (#A-21447, ThermoFisher, 1:1000), AF 647-conjugated chicken anti-rat (#A-21472, ThermoFisher, 1:1000), AF 488-conjugated donkey anti-rabbit (#A-21206, ThermoFisher, 1:1000), AF 568-conjugated donkey anti-mouse (#A10037, ThermoFisher, 1:1000), AF 647-conjugated goat-anti rabbit (#A-21443, ThermoFisher, 1:1000) and AF 488-conjugated donkey anti-goat (#A-11055, ThermoFisher, 1:1000). The list of primary and secondary antibodies and dilutions used for IF is in Supplementary Data 5.

Immunofluorescence images were acquired with a ZEISS LSM880 confocal microscope with 20×, 40×, or 63× oil immersion objectives or with a NanoZoomer S60 microscope with a 40× objective. ZEN Blue software was used for image acquisition. Fiji/ImageJ software was used for image processing and analysis. For image analysis, the images were stacked to maximal projections, and immunofluorescent channels were split. A binary mask was then created using a watershed function in the DAPI channel, allowing for the identification of individual nuclei. The mean gray value intensity of channels was measured and summed. The fluorescent intensities are indicated in arbitrary units. On average, >100 cells were analyzed for in vitro studies. For in vivo studies, five fields were imaged per tumor. Imaris software v9 (Oxford Instruments) was used to create the 3D tumor reconstruction from thick tumor sections (40 µm). Z stack images of tumor sections were acquired using the Zeiss LSM880 confocal laser scanning microscope. The 3D volume was recreated by the surface rendering method in Imaris.

Immunohistochemical staining was performed by the laboratory of pathology in the Department of Biochemistry, UNIL, as previously

described (Ma et al. 2021). Slides were scanned using a NanoZoomer S60 microscope with a 20× objective. Ndp.View2 and Fiji/ImageJ software were used for the acquisition and processing of images.

## Immunoblotting

For whole cell extracts, cells were lysed using boiling LDS buffer (2% SDS, 50 mM Tris/HCl (pH 7.4) supplemented with 1 mM PMSF, 1 mM Na3VO4, and 10 mM NaF. Total protein content was quantified with a BCA assay (Thermo Fisher Scientific). Equal amounts (20-50 µg) of proteins were subjected to 10% SDS–PAGE followed by immunoblot analysis.

Primary antibodies used were anti-Androgen Receptor (#5153, Cell Signaling Technology, 1:1000), anti-EGFR (#4267, Cell Signaling Technology, 1:1000), anti-PAI-1 (#19773; Novus Biologicals, 1:200), anti-Axl (#3269, Cell Signaling Technology, 1:1000), anti-GAPDH (#sc-25778, Santa Cruz Biotechnology, 1:10,000), anti-Histone H3 (#4499, Cell Signaling Technology, 1:10,000). The secondary antibody was anti-Rabbit IgG (H + L), HRP Conjugate (#W401B, Promega, 1:10,000).

Cytoplasmic and nuclear fractions were obtained using NE-PER™ Nuclear and Cytoplasmic Extraction Reagents (Thermo Fisher Scientific) according to the manufacturer's instructions. In brief, $1 \times 10^6$ cells were lysed using a cytoplasmic extraction reagent for 10 min. Following, intact nuclei were pelleted down at maximum speed for 5 min. Nuclear extraction was then performed using RIPA buffer (Thermo Fisher Scientific) supplemented with a protease inhibitor cocktail (Roche) for 20 min and then centrifuged at maximum speed for 10 min.

Protein content was quantified with a BCA assay (Thermo Fisher Scientific). Equal amounts (20-30 µg) of proteins were subjected to 10% SDS–PAGE followed by immunoblot analysis. All membranes were sequentially probed with different antibodies as indicated in the figure legends. Super Signal West Pico PLUS Chemiluminescent Substrate (Thermo Fisher Scientific) was used for signal detection. Full details of antibodies used in this study are provided in Supplementary Data 5.

## Real-time quantitative PCR (RT-qPCR)

Total mRNA was extracted using TRIzol according to the manufacturer's instructions, followed by cDNA synthesis using the RevertAid H Minus Reverse Transcriptase (Thermo Fisher Scientific). RT-qPCR was performed using SYBR Fast qPCR Master Mix (Kapa Biosystems) on a Light Cycler 480 (Roche). The relative quantification (RQ) and expression of each mRNA were calculated using the comparative Ct method. All samples were run in technical triplicates and were normalized to an endogenous control, *RPLPO*. A full list of primers used in the study is provided in Supplementary Data 5.

## Transcriptomics and bioinformatics analysis

For transcriptomics analysis of BRAFi-resistant and parental A375, M160915, M121224, UACC903, and WM9 melanoma cells, RNA was extracted from 500,000 melanoma cells as described below. For acute DAB treatments, A375, M14, and WM9 melanoma cells infected with an *AR* overexpression lentivirus versus LacZ expressing control virus were treated with Dabrafenib (0.5 µM) versus DMSO vehicle for 48 hours. Following, RNA was extracted using the Direct-zol RNA MiniPrep kit (Zymo Research) coupled with DNase treatment according to the manufacturer's instructions. The RNA quality was first evaluated using Agilent 2100 Bioanalyzer® (Agilent Technologies, USA). Transcriptomic analysis was performed using Clariom™ D GeneChip array hybridization (Thermo Fisher Scientific). Single-strand cDNA preparation, labeling, and hybridization were performed in accordance with Affymetrix protocols at the iGE3 Genomics Platform, University of Geneva (Geneva, Switzerland). Data obtained (CELL files) were summarized using the RMA function in the R package oligo with background correction and quantile normalization. Gene IDs were mapped using the Chip-

annotation package clariomdhumantranscriptcluster.db. The R package "limma" was used for gene differential expression analysis, followed by multiple testing correction by the Benjamini-Hochberg procedure. The cutoffs for the Dabrafenib treatment signatures (Dabrafenib CNTRL vs DMSO CNTRL and Dabrafenib AR OE vs AR OE CNTRL) were FC > 1.5, and adj-p < 0.01, yielding 360 up- and 360 down-regulated genes for CNTRL Dabrafenib-treated cells, and 199 up- and 344 down-regulated genes for AR OE Dabrafenib-treated cells. The cutoff for the AR OE signature (AR OE CNTRL vs DMSO CNTRL) was FC > 2.0, and p-value < 0.05, yielding 48 up- and 39 down-regulated genes (GSE232697).

Gene ontology (GO) enrichment analysis was performed on the differentially expressed genes with the fold change cutoff value of 2.0 using the Enrichr. Gene Ontology and Pathway Classification System to identify the enriched biological processes.

Gene set enrichment analysis (GSEA) for GeneChip microarray data was conducted using GSEA software using default parameters. Curated gene sets were obtained from the various sources indicated in the legends for Figs. 3 and 4: (i) the Molecular Signatures Database (MSigDB version 5.2, www.broadinstitute.org/gsea/msigdb/); (ii) previously published melanoma-specific signatures[21,23,48]. A list of enriched pathways is provided in Supplementary Data S1,2,3. To assemble the AR signaling pathways collection, we utilized the Molecular Signatures Database and filtered for signatures of min. size of 20 genes and max. size of 500 genes. The resulting 16 gene signatures plus the established prostate-specific signature of AR activity[46] were used to perform GSEA and leading-edge analysis using software default parameters against the transcriptional profiles of BRAFi-resistant melanoma cells. A list of enriched pathways and leading-edge genes is provided in Supplementary Data S1.

For AR overexpression score, CELL files were summarized using the RMA function in the R package oligo with background correction and quantile normalization. Gene IDs were mapped using the Chip-annotation package clariomdhumantranscriptcluster.db and differential expression analysis was performed with limma, using the formula -treatment + cell_line. The treatment referred to the comparison DMSO versus AR OE.

Signature score analysis of single-cell RNA-seq profiles was performed starting from single cell RNA-seq data (GEO # GSE116237) filtering for cells with more than 1000 gene counts and genes detected in more than 3 cells. Further filtering was omitted as it has already been done by the authors of the dataset[21]. Ensembl IDs were mapped into gene symbols using biomaRt[49] and count data were summed together when multiple IDs mapped to the same symbol. Library normalization, log transformation and further downstream analysis were performed using Seurat v4[50]. Signature scores were calculated using AUCell[51] and significance between scores or individual gene expressions were calculated using Welch's t-test[52]. Gene sets were downloaded from the Molecular Signatures Database[22].

Correlation analysis between AR, EGFR, PAI-1, and AXL expression levels was calculated on 472 melanoma samples from the TCGA project (TCGA Firehose Legacy, February 2022) with the corrplot 0.92 package, using the Spearman's correlation method.

Survival analysis was based on the melanoma TCGA dataset and calculation of the optimal cutpoint for continuous variables (log2Expression value = 3.08) from the maximally selected rank statistics from the 'maxstat' R package.

## ChIP-seq and bioinformatic analysis

Cells were cross-linked with 1% formaldehyde for 10 min at RT. The cross-linking reaction was then stopped by adding glycine (125 mM). Cells were washed with PBS and pelleted (400 x g, 5 min). The cells were lysed, and the chromatin was prepared using the Chromatin EasyShear Kit - High SDS (Diagenode). Sonication was performed using the E220 focused ultrasonicator (Covaris). For the ChIP (chromatin immunoprecipitation) assay, the iDeal ChIP-seq kit (Diagenode) was

used according to the manufacturer's instructions. The samples were pre-cleared with the beads provided in the kit and then incubated overnight at 4 °C with 5ug of anti-Androgen Receptor antibody (PG-21, Cat. No. 06-680, Sigma). The antibody-chromatin complexes were pulled down using protein A-coated magnetic beads. Elution of the complexes was carried out following the manufacturer's instructions, and the chromatin was quantified using the Qubit Fluorometric Quantification Kit (Thermo Scientific). For library preparation, 10 ng of DNA was used with the NEBNext® ChIP-Seq Library Prep Reagent Set for Illumina. Sequencing was performed at the Novogene facility using the Novaseq 6000 platform. The resulting FASTA files were aligned using the Burrows-Wheeler Aligner (BWA) 70, and MACS2 software was used to identify narrow peaks with a q-value cut-off of 0.05. The Integrative Genome Viewer (IGV) software was used to generate graphical representations of the ChIP-seq peaks. Peaks were annotated and merged using the annotatePeaks.pl and mergepeaks.pl functions from the HOMER software.

RP score analysis was performed using the Cistrome DB toolkit (http://dbtoolkit.cistrome.org/), a comprehensive bioinformatics tool for the analysis of ChIP-seq data, using default parameters. The Cistrome DB toolkit was utilized to identify the *overlap ratios* between transcription factors and chromatin regulators with the differential AR OE and DAB peak sets. To identify transcription factors binding *AR* ChIP-seq peaks on the *AR gene*, the individual intervals or genome coordinates of the four identified AR ChIP-seq peaks were analyzed in Cistrome, and the transcription factors that exhibited binding overlap across all four peaks were identified (**GSE232697**).

### In vivo studies

NOD *scid* gamma (NSG) mice, (NOD.Cg-Prkdcscid Il2rgtm1Wjl/SzJ; 6-8-week-old males), were obtained from the Jackson Laboratory. BRAFi resistant human A375 melanoma cells (A375BR) were pretreated with AZD3514 (10 μM) or DMSO control for 12 hours prior to injection into mice. Cells ($1 \times 10^6$ per injection, in Matrigel (Corning), 70 μl) were injected intradermally in parallel into the left and right flanks of mice with 29-gauge syringes. Mice were sacrificed and Matrigel nodules were retrieved for tissue analysis 10 days after injection.

C57BL/6JRj mice (6-8-week-old males) were obtained from Jackson Laboratory. BRAFi resistant murine melanoma cells (YUMM1.7BR) were pretreated with AZD3514 (10 μM), ARCC4 (1 μM), or DMSO control for 12 h prior to injection. $2 \times 10^6$ melanoma cells per condition were injected with Matrigel (Corning) (70 μl per injection) intradermally in parallel into the left and right side of mice with 29-gauge syringes. Mice were sacrificed and Matrigel nodules were retrieved 14 days after injection. Tumors were measured post-extraction using calipers.

In the in vivo systemic treatment experiments with AZD3514, ARCC4, and Enzalutamide, male NOD SCID mice aged 6-8 weeks were intradermally injected with 200,000 YUMM1.7BR melanoma cells suspended in Matrigel solution on their back skin. Three days after injection, the mice were divided into different groups. Each group consisted of five mice and received the following treatments by oral gavage for a period of 14 consecutive days: AZD3514 (50 mg/kg), ARCC4 (30 mg/kg), Enzalutamide (50 mg/kg), or Corn Oil (Sigma) as the vehicle control. The volume of the tumors was measured every two days using digital calipers, and the tumor volumes were calculated using the formula ($LxW^2x0.5$). The maximal tumor size permitted by the approved protocol was 20 mm in any direction, which was not exceeded in any experiment. All mice were euthanized by carbon dioxide inhalation. No animal died or was excluded from the experiments. Mice were housed in four per cage on a 12-hour dark/light cycle, with a constant ambient temperature of 65-70 F and 40% humidity. All the mice were randomly allocated to the experimental groups and group-housed. Throughout the study, humane endpoints were implemented to ensure the well-being of the mice. All mice were housed in the animal facility of the University of Lausanne.

### Statistical analysis and reproducibility

Data are shown as mean ± SEM or mean ± SD, as indicated in the legends. All western blots were performed at least twice. When a representative image is shown, the number of samples and conditions analyzed are provided in the figure legend. Detailed information on the statistical methods applied for each experiment is in the corresponding figure legends. Statistical difference between the two groups was determined using Student's *t*-test unless otherwise mentioned. For comparisons among more than two groups, a one-way analysis of variance (ANOVA) followed by Bonferroni's correction was used. For longitudinal data, Spearman's correlation was used to infer significance between the experimental treatment arms.

For tumorigenicity assays, individual animal variability issue was minimized by contralateral injections in the same animals under control versus experimental conditions. No statistical method was used to predetermine sample size in animal experiments and no exclusion criteria were adopted for studies and sample collection. No exclusion criteria were adopted for animal studies or sample collection. No randomization was used, and the researchers involved in the study were not blinded during sample obtainment or data analysis. All statistical tests were performed using GraphPad Prism 9 (GraphPad Software, Inc.). Information for all software and algorithms used is in Supplementary Data 5.

### Reporting summary

Further information on research design is available in the Nature Portfolio Reporting Summary linked to this article.

## Data availability

The transcriptomic and ChIP-seq data generated in this study have been deposited in GEO (Gene Expression Omnibus), NCBI with an accession number GSE232697 (https://www.ncbi.nlm.nih.gov/geo/query.acc.cgi?acc=GSE199405). There is no restriction on data availability. No code was developed in this study. Standard pipelines and open-source R packages were used. Information for PCR oligos, vectors, compounds, and antibodies used are in Supplementary Data 5. Source data underlying Figs. 1a–h, 2a–h, 3a–g, 4a–f, 5a–f, 6a–i, 7a–f, 8a–g 9a-m, and Supplementary Figs. S1a–c, S1d–h, S1j-m, S2a, b, S2d, e, S3d, S4a, b, S5b, S6b–e, S7b–e, S9b is provided with this paper. Source data are provided in this paper.

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

## Acknowledgements

We are grateful to Meenhard Herlyn for providing WM115, WM9, WM989, WM983A, WM9, WM9BR, UACC903, and UACC903BR melanoma cell lines used in the study. We thank Ping-Chih Ho for a gift of murine YUMM1.7 melanoma cells. We are grateful to Luigi Mazzeo, Jovan Isma, and An Buckinx for stimulating discussions and critical reading of the manuscript. We thank the URPP in Translational Cancer Research Biobank at the University of Zürich for access to the melanoma cell lines and patient samples. This work was supported by grants from the Swiss Cancer Research Foundation (project number: KFS-4709-02-2019) and the National Institutes of Health (R01AR078374; R01AR039190; content not necessarily representing the official views of NIH). MKY is supported by the European Union's Horizon 2020 Research and Innovation Program under Marie Skłodowska-Curie Grant Agreement No 859860. MPL is a member of the SKINTEGRITY.CH collaborative research program.

## Author contributions

Author initials: Soumitra Ghosh = SG; Sandro Goruppi = SGO. A.S., S.G., T.P., B.T., M.M., and S.G.O. performed experiments and/or analyzed the results with G.P.D. M.K.Y., P.O., and G.C. performed the bioinformatic analysis, and M.P.L. provided the clinical samples and critical feedback. A.S. and G.P.D. designed the study and wrote the manuscript.

## Competing interests

The authors declare no competing interests.
