## [Peer Review file · Nature Communications]

REVIEWER COMMENTS

Reviewer #1 (Remarks to the Author): expertise in transcriptomics in melanoma

In this study, Samarkina et al. investigated the role of androgen receptor signaling in melanoma response to targeted BRAF inhibitors. Using in culture treatment of primary melanoma cells and patient samples from before and after therapy, they show that AR expression is increased after BRAF inhibitors treatment. By comparing the transcriptional profiles of cell lines with/without AR overexpression after treatment they show that AR expression contributes towards reprogramming of melanoma subpopulations into resistant cells by inducing transcriptional changes indicating suppression of immunomodulatory, antigen-presentation and pro-apoptotic pathways. Treatment with AR inhibitors induces infiltration of CD8 T cells and suppresses proliferation and tumorigenicity of resistant cells.

The study shows a very interesting link between AR signaling and resistance to targeted therapy and has potential clinical significance. What was missing for me was some deeper mechanistic studies of how exactly AR signaling induces resistance and why AR expression gets upregulated after therapy and some further confirmation that there isn't a pre-existing potentially resistant subpopulation of melanoma cells with higher AR expression.

Specific comments:

- Is AR expression homogenous across different melanoma subpopulations in untreated melanoma patient samples or is it upregulated before treatment in a specific subpopulation? Publicly available single-cell RNA-seq data from melanoma patients would help with this. AXL high subpopulation has been identified also in treatment naive patients (Tirosh et al, 2016). If this is indeed an adaptive resistance mechanism, it would be very useful to get more insights into how BRAF inhibitors induce upregulation of AR.

- Related to this, it would be good to get deeper mechanistic insights into the link between AR and EGFR and TGF- β , potentially further analysis such as inference of gene regulatory network of the scRNA-seq data from the PDX could help or alternatively transcriptional analysis of the cell lines/mouse models treated with a BRAFi and AR inhibitors..

- Figure 3A, it would be very useful to name some of the relevant DE genes with high/low log₂ fold change on the volcano plots.

Reviewer #2 (Remarks to the Author): expert in melanoma mouse models

In this study, Samarkina and co-authors discovered that the Androgen Receptor (AR), besides being involved in cell proliferation and tumorigenesis, has an important role in BRAF inhibitor resistance of melanoma cells. MAPK-targeted therapies, such as BRAFi, have provided a clinical benefit not previously observed in metastatic melanoma of BRAF/RAS-mutated patients. However, despite all these efforts, melanoma treatment remains challenging because of the development of resistance to therapy. Here the authors demonstrate that AR increases upon BRAFi treatment and is upregulated in BRAFi-resistant melanoma cells. This is associated to the expression of EGFR/SERPINE1 genes both in melanoma cell lines and in publicly available datasets of melanoma patients. They further demonstrate that the pharmacological inhibition of AR in BRAF resistant cells suppresses tumor growth and induces CD8 T-cells infiltration/MHCI expression of BRAFi resistant cells-derived tumors in xenografts/syngeneic mouse models.

Despite being very interesting and of high significance in the field, it is still quite preliminary and further data are needed to implement it. My major concern relies on the fact that the molecular mechanism(s) underlying the AR-mediated resistance phenotype in melanoma cells was not deeply investigated. An extensive bioinformatic analysis (both in melanoma cells and public datasets) has been done to identify the functional pathways/gene signatures modulated by AR OE and in BRAFi-resistant cells, but none of them has been experimentally evaluated. Another concern is the in vivo studies, which are quite limited. Experiments demonstrating that AR is upregulated also in vivo upon exposure to BRAFi and that tumor-bearing mice become resistant to BRAFi upon AR OE should be addressed.

Here are my comments, point-by-point:

- A general concern is that the statistics of many qPCRs analyses is missing, as for Fig. 1A,E,F, Fig. 5A and suppl. Fig 1A-D.
- Moreover, in Fig. 1A, why is there not any protein band of AR at 1uM of treatment? While the RNA is already upregulated at 0.5uM doses?
- Densitometry is required in Fig. 1B,E
- The statistics is missing from the heatmap shown in Fig. 1D
- With a closer look at the original values of the first 3 genes from suppl. Fig 2A, mentioned in the text, they are actually pretty similar (DMSO/DAB). A qPCR should be done at least for the 3 genes in the different melanoma cell lines upon DAB treatment. Furthermore, a more in-depth examination of the role of these factors in AR regulation in BRAFi resistant/parental cells is warranted.
- The statistics is also missing in Fig. 2A.
- Why does Fig. 2E show a higher number of colonies in DMSO/CTR conditions, compared to Fig. 2D where the same A375 cells were used? The difference with AR OE is indeed quite low (compared to Fig. 2D).
- It should be better clarified in the text lines 161, 164, in relation to Fig.3 C-E. It is not so obvious from the text that the second GSEA plot (in C,D, and E) compares DAB ctr versus DAB AR OE.
- Fig. 5B should include the densitometry analysis.
- The in vitro experiments in Fig.5 should be repeated upon genetic inhibition of AR, at least in some of the cell lines. All the human BRAFi-resistant cell lines used in the cell viability/cell death tests should be analysed for EGFR/SERPINE1 activation; more

consistency with the different cell lines used in the different assays is required. In Fig. 5G, the untreated cells should also be included.

- The in vivo experiments in Fig. 6A-E should include the tumor kinetics analysis to see how the tumor growth of AZD/ARCC4 treated tumors decreases over time. Also, a more thoughtful analysis of the immune cell populations in the TME of YUMM1.7BR-derived tumors should be addressed. Flow cytometry analysis of at least T-cells populations should be included to confirm the IF data. Another weak point is that no characterization in terms of AR expression and cell viability upon BRAFi of YUMM1.7 BR cells has been done. It should be included to complete this part.

Moreover, in both xenograft/syngeneic models further analyses showing the status of the BRAFi-resistant cells in terms of EGFR/SERPINE activation are missing.

- In addition, if EGFR/TGFb pathways are hyperactivated once AR increases upon BRAFi exposure, can they be somehow targeted to rescue the AR-mediated resistance phenotype?

Reviewer #3 (Remarks to the Author): expert in BRAF inhibitors in melanoma

This study reports that the Androgen Receptor (AR) is a mediator of resistance to BRAF inhibitors (BRAFi) in BRAF mutant melanoma. It is shown that cells grown in high concentrations of BRAFi consistently upregulate AR, but not other common melanoma resistance genes. AR is upregulated within 48 hours of BRAFi or MEKi but is not upregulated by inhibitors of several other pathways. Tumours from patients who developed resistant also upregulate AR, and bioinformatic analysis shows that transcription factors (TF) known to upregulate AR are also increased, although the role of these TF is not tested directly. Cell growth when AR is over-expressed is less sensitive to BRAFi and the cells are more resistant to apoptosis. Bioinformatic analysis reveals that within 48 hours of BRAFi, pathways that control cell cycle are downregulated, but interferon and antigen presentation are upregulated. In cells over-expressing AR, Serpine and EGFR pathway components are upregulated, and these pathways correlate to distinct resistant sub-populations. High expression of AR/EGFR are associated with reduced survival. EGFR and Serpine are upregulated in cells when AR is over-expressed or the cells are grown to be resistant to BRAFi and the growth of the resistant cells is sensitive to AR antagonists. AR antagonists suppress the emergence of BRAFi colonies in vitro. Finally, pre-treatment of cells with AR antagonists suppresses human xenograft tumour growth and increased CD8+ T cell infiltration in BRAF mutant mouse allografts.

The data support the conclusions, but several weaknesses dampen enthusiasm for publication.

General comment. There is a lack of consistency between the use of cell lines and techniques and no single cell line is carried as an exemplar throughout all the study to give consistency of narrative and approaches. This makes it difficult to compare between cell lines and experiments, resulting in a broken narrative and making it difficult to be fully convinced by the data. Specifically:

1. Figure 1. The premise of this figure is that cells grown long-term in high concentrations of BRAFi (dabrafenib/DAB) become resistant to the inhibitor and increase AR expression.

However, the resistant cells are not characterised to provide the reader with a baseline from which to understand the data.

a. Cell growth against DAB concentration curves should be provided.

b. ERK activity against DAB concentration curves should be provided.

c. Panel E. It is curious that AR levels rise so quickly (within 48 hours), because most of these cells in these cultures will eventually die, which suggests that the AR elevation is insufficient to save most of the cells in the culture. The authors should test if the increase in AR in this time frame is restricted to a small population of cells that will survive and emerge as resistant. If all the cells express AR, how do the authors consolidate that observation with their model?

d. Please comment on the high concentrations of DAB used to drive resistance and how the concentration relates to the sensitivity of the parental lines.

2. Figure 2. Please provide western blots to confirm that AR over-expression compared to the parental cells and to the cells driven to resistance by growth in increasing concentrations of DAB.

3. Whilst Figures 3 and 4 present some interesting bioinformatic data, they do not carry the narrative much further, as largely the observations are not followed up. Most of this data could go to supplementary with just the key observations being presented in a reduced Figure 3.

4. Figure 5.

a. Panel C. Please provide western blots to accompany the RT-PCR data.

b. Panels E-G. Please demonstrate specificity by comparing the impact of AR antagonists on the growth of non-resistant parental cells.

c. The narrative would be significantly strengthened if it was shown that EGFR and Serpine modulated resistance to BRAFi downstream of AR (i.e. are the cells addicted to these proteins?).

5. Figure 6.

a. While the demonstration that pre-treatment of cells with AR antagonists reduces tumour growth in mice is of some interest, it is not surprising or particularly informative. The authors show in Figure 5F that AR antagonists cause cell death in vitro, and it is unsurprising that dying cells are less able to form tumours than healthy cell controls. The appropriate experiments to provide pre-clinical context are:

i. Demonstrate that AR antagonists suppress the growth of BRAFi resistant or AR-over-expressing BRAF mutant cells growing as established tumours in mice.

ii. Demonstrate that AR antagonists cooperate with BRAFi to suppress the emergence of resistant clones in parental cells growing as established tumours in mice.

b. Panels C-F. The functional relevance of MHC I surface expression and increased CD8+ T cell infiltration is unclear and untested. It could simply reflect the fact that the cells are dying when injected, and it is the cell death rather than AR antagonist specific effects that are causing the infiltration. The appropriate experiments are:

i. Provide (brief) evidence that YUMM1.7BR cells behave in the same way as human cell lines in vitro:

1. Is AR elevated in YUMM1.7BR cells compared to sensitive parental YUMM1.7 cells?

2. Are YUMM1.7BR cells sensitive to AR antagonists in vitro compared to the parental cells?

ii. Show that the increase of the immune cell infiltration occurs when mice bearing established tumours are treated with AR antagonists.

iii. Demonstrate specificity by showing T cell infiltration does not occur when BRAFi sensitive parental YUMM1.7 cells are treated with AR antagonists in mice.

Reviewer #4 (Remarks to the Author): expert in androgen receptor signalling

This manuscript by Samarkina et al., is focused on unravelling a novel mechanism of resistance in response to BRAF/MEK inhibitors in melanoma patients. In this study, the authors have shown that expression of androgen receptor (AR) is significantly elevated in melanoma cells exposed to BRAF/MEK inhibitors. Increased AR expression is important for maintaining the tumorigenicity in these cells and is accompanied with high levels EGFR and SERPINE1 expression which could confer a tumor survival advantage. In addition, blocking AR signaling in BRAFi-resistant cells leads to increased expression MHC I expression and CD8+ T cells infiltration. This body of work is very important in setting a possible precedent for androgen receptor blockade as a therapy against BRAFi/MEKi-resistance in melanoma patients. However, this study seems more correlative in nature and does not dwell deeper into the mechanism behind AR upregulation. Moreover, this study relies much on cell line data rather than clinical melanoma data. Attention to the following points can further help improve the quality of the manuscript.

Major Comments:

1. The author should include the recent study by Vellano et al., Nature, 2022 in the introduction.

2. In several bar graphs, error bars are missing. Please add the error bars.

3. Some of the figure legends are overly descriptive. The figure legends should be more succinct, and the information about the experiment details should be explained in the methods section.

4. Figure 1, apart from checking AR expression, the authors should also check the expression of AR regulated genes from prostate cancer. This will also help in understanding the distinct mechanism of regulation mediated by AR in melanoma.

5. The authors should perform some experiments with androgens to confirm the ligand-dependent or ligand independent activity of AR in melanoma cells.

6. In Supplementary Figure 1D, protein level for AR in other melanoma cell lines would strengthen the manuscript.

7. The authors have established the connection of FOXO3, CREB1 and SP1 with AR expression based on the literature and correlative expression patterns. The authors should also perform some genetic perturbations in either parental cells (overexpression) or BRAF resistant cells (knockdown) to confirm whether these proteins are regulating AR expression.

8. In Figure 2, author should perform additional functional assays (such as invasion, migration or tumorsphere formation assays) to assess the role of AR overexpression in melanoma cells in response to BRAF inhibition.

9. In Figure 3, the authors should also perform GSEA analysis for androgen signaling to check for the expression of conventional androgen-regulated genes.

10. In supplementary figure 4, authors have used the AR inhibitors AZD3514 (10 μ M) or ARCC4 (1 μ M). Is there any particular reason why authors did not use anti-androgens such as enzalutamide or apalutamide, that are frequently used to inhibit AR activity in prostate cancer?

Minor Comments:

1. In Figure panel 1D and 3B, the authors should consider different colours for the heat map to distinguish between the high and low gene expression changes.

REVIEWER COMMENTS

Reviewer #1 (Remarks to the Author): expertise in transcriptomics in melanoma

In this study, Samarkina et al. investigated the role of androgen receptor signaling in melanoma response to targeted BRAF inhibitors. Using in culture treatment of primary melanoma cells and patient samples from before and after therapy, they show that AR expression is increased after BRAF inhibitors treatment. By comparing the transcriptional profiles of cell lines with/without AR overexpression after treatment they show that AR expression contributes towards reprogramming of melanoma subpopulations into resistant cells by inducing transcriptional changes indicating suppression of immunomodulatory, antigen-presentation and pro-apoptotic pathways. Treatment with AR inhibitors induces infiltration of CD8 T cells and suppresses proliferation and tumorigenicity of resistant cells.

The study shows a very interesting link between AR signaling and resistance to targeted therapy and has potential clinical significance. What was missing for me was some deeper mechanistic studies of how exactly AR signaling induces resistance and why AR expression gets upregulated after therapy and some further confirmation that there isn't a pre-existing potentially resistant subpopulation of melanoma cells with higher AR expression.

We thank the reviewer for finding our work of interest and the recommendations for improvement. We have addressed the three primary concerns as follows :

1) mechanistic studies of how AR signaling induces BRAFi resistance

To investigate the underlying mechanisms responsible for the induction of long-term BRAFi resistance, we had already compared the transcriptional profiles of three melanoma cell lines with and without AR overexpression. As explained in the text (p. 11, top), we have expanded this work, by ChIP-seq analysis with anti-AR antibodies of A375 melanoma cells with and without AR overexpression (Fig. 6d-g). We then compared these profiles with the ChIP-seq profiles of AR in the same cells after early treatment with BRAFi (DAB). Our rationale was that short-term exposure of melanoma cells to BRAF inhibitors leads to the significant induction of AR expression, accompanied by gene expression changes that, however, are insufficient to cause BRAFi resistance. The following steps were undertaken:

a) Initially, we focused on the *SERPINE1* gene, a known TGF- β target with pro-tumorigenic functions. This gene exhibited the highest upregulation in melanoma cells overexpressing AR and was also upregulated in cells selected for BRAFi resistance, but not in the parental lines following short-term BRAFi exposure (Fig 6A, Supp. Fig 5B, Suppl. Table 2-3). We observed significantly elevated AR binding peaks at the promoter, first exon, and second exon regions of *SERPINE1* exclusively in the AR-overexpressing cells, not in cells treated with DAB for 48 hours (Fig. 6d).

b) Through a global analysis of the ChIP-seq profiles, we identified a notable increase in AR binding near the transcription start sites (TSS) of several genes in both AR-overexpressing A375 melanoma cells and the same cells treated with DAB for 48 hours. However, there was only partial overlap between AR target genes that were upregulated in these two conditions. One group of genes (65) was bound by AR and induced in both AR-overexpressing and DAB-treated cells, while another group (202) was selectively bound by AR and induced solely in the AR-overexpressing cells (Fig. 6e).

c) To identify known transcription factors and chromatin regulators that bind to sites overlapping with selective AR binding in AR-overexpressing cells, and potentially collaborate with AR function, we performed an unbiased analysis of publicly available ChIP-seq profiles using the Cistrome Data Browser and toolkit (<http://dbtoolkit.cistrome.org/>). Among the top-ranked transcription factors with a high score of overlap with the identified AR binding peaks across various cell types were c-Myc, E2F1, and SP1. Additionally, BRD4, a protein involved in chromatin organization and transcription, along with a histone demethylase (KDB2B), a chromodomain helicase (CHD1), and a RNA polymerase subunit (POLR2A), were among the top-ranked proteins (Fig. 6f).

d) Gene ontology analysis of genes selectively bound by AR and upregulated in AR-overexpressing cells, as well as broader gene set enrichment analysis (GSEA), revealed a significant enrichment of genes related to TGF- β and EGFR signaling, which have previously been linked to the acquisition of BRAFi resistance (Fig. 6 g, h).

e) To assess whether increased TGF- β and/or EGFR signaling could account for the DAB resistance resulting from persistent AR expression, we treated cells with inhibitors targeting these pathways individually and in combination. Treatment with TGF- β and/or EGFR inhibitors did not affect the proliferation of A375 parental cells, nor did it enhance the suppressive effects of DAB treatment (Fig. 6i). Similarly, the proliferation of AR-overexpressing cells was unaffected by treatment with TGF- β and/or EGFR inhibitors alone. However, when these inhibitors were administered concomitantly with DAB, the sustained proliferation of AR-overexpressing cells was significantly reduced, with stronger suppressive effects observed when the two inhibitors were combined (Fig. 6i).

In conclusion, sustained AR expression leads to the selective binding and upregulation of a set of genes involved in the TGF β - and EGFR-signaling pathways. Targeting these pathways effectively suppresses AR-induced BRAFi resistance. As considered in the discussion (p. 19, line 445-450), the selective targeting of these genes by AR in cells with persistent expression of the gene, as opposed to cells with transient AR upregulation due to DAB treatment, may result from a complex cascade of events triggered by AR upregulation. These events likely involve various direct and indirect mechanisms that determine chromatin configuration and accessibility. Alternatively, a preexisting subpopulation with a specific chromatin configuration that responds differently to increased AR expression may be selectively amplified over time.

2) why AR expression gets upregulated after therapy

The upregulation of AR can occur either as a result of chronic treatment with BRAFi or as part of an acute response. In our original paper, we demonstrated that significant induction of AR expression took place in various primary and established melanoma cells as early as 48 hours after treatment with DAB and other BRAF and MEK inhibitors (Fig. 2a). Conversely, inhibitors targeting other key signaling pathways, such as NF- κ B, STAT3, and AP-1, did not elicit a similar effect (Suppl. Fig. 2a).

To delve deeper into the underlying mechanisms, we focused our investigation on A375 melanoma cells. Through immunofluorescence (IF) analysis, we observed an increase in nuclear AR protein levels just two hours after DAB treatment, with a more pronounced increase at 24 and 48 hours (Fig. 2f, Suppl. Fig. 2c). Subsequent cell fractionation and immunoblot analysis confirmed enhanced nuclear localization of the

AR protein after 48 hours of DAB treatment, similar to what was found in BRAFi-resistant cells (Fig 2g, h; Suppl. Fig 2d, e).

These findings led us to explore the possibility that AR binds to chromatin early in the course of DAB treatment and upregulates genes of interest, potentially including itself. By ChIP-seq analysis with anti-AR antibodies, we observed a marked increase in AR binding to genomic regions encompassing the transcription start sites (TSS) of genes in A375 melanoma cells after 48 hours of DAB treatment (Fig. 3A). Integrating the ChIP-seq profiles with transcriptomic profiles of A375 cells and two other melanoma lines with or without DAB treatment for 48 hours, we identified 123 genes as direct targets of AR induction by DAB treatment (Suppl Table 2). Notably, the *AR* gene itself ranked among the top five target genes (Fig. 3c), with a peak of markedly induced AR binding at its TSS site.

It is well established that AR can regulate its own expression through a positive feedback loop. As indicated in the text (p. 7, line 161), to functionally assess whether AR itself is involved in its upregulation in response to DAB treatment, we co-administered two AR-degrading compounds, known as proteolysis-targeting chimeras (PROTACs). We observed that the induction of AR expression by DAB treatment was completely suppressed by these compounds, as well as by an inhibitor targeting the AP1 transcription complex (Fig. 3F, G). The AP1 complex is known to synergize with AR in the control of gene expression and can be induced by BRAFi treatment as part of a compensatory mechanism. Induction of *AR* expression by BRAFi was not reduced by concomitant treatment with inhibitors of the EGFR and TGF β pathways (Fig. 3f), which in a later part of the work we found to be responsible for AR-induced BRAFi resistance.

In conclusion, our findings demonstrate that the early induction of AR expression by BRAFi treatment involves a positive feedback loop, with the AR protein translocating into the nucleus and binding to the regulatory region of the AR promoter. As highlighted in the discussion (p. 18, line 421), this process may involve post-transcriptional modifications of the AR protein and/or its dissociation from cytoplasmic retaining proteins. Further dedicated studies will be necessary to systematically investigate these possibilities.

3) further confirmation that there isn't a pre-existing potentially resistant subpopulation of melanoma cells with higher AR expression

In accordance with the aforementioned biochemical evidence, our immunofluorescence analysis demonstrates induction of AR expression in the vast majority of melanoma cells at early times of DAB treatment (Fig. 2f). This observation does not dismiss the possibility of pre-existing subpopulations with heightened *AR* expression or, as pointed out in the discussion (p. 19, line 452), subpopulations exhibiting distinct chromatin configurations that respond differentially to increased *AR* expression. This may explain the previously discussed differences in AR chromatin binding profiles, along with the associated alterations in gene expression, between melanoma cells subjected to short-term DAB treatment and those with sustained elevation of *AR* expression.

Specific comments:

- *Is AR expression homogenous across different melanoma subpopulations in untreated melanoma patient samples or is it upregulated before treatment in a specific*

subpopulation? Publicly available single-cell RNA-seq data from melanoma patients would help with this. AXL high subpopulation has been identified also in treatment naive patients (Tirosh et al, 2016). If this is indeed an adaptive resistance mechanism, it would be very useful to get more insights into how BRAF inhibitors induce upregulation of AR.

We thank the reviewer for the interesting questions. In our recent study, we demonstrated the heterogeneous expression of AR in melanoma cells, both at the single-cell level within a lesion and across lesions at different disease stages (Ma et al., JEM 2021). In accordance with the reviewer's suggestion, we conducted an analysis on the single-cell dataset of melanoma cells obtained from patients prior to BRAFi treatment, as provided by Tirosh et al. in 2016. This analysis involved the gene signature we established from melanoma cells with AR overexpression. Consistent with the single cell analysis of the PDX model of Rambow et al. that we had already shown (Fig. 7a), we have found a high AR signature score in tumor cell populations characterized by an “elevated AXL program” identified by Tirosh et al, 2016 and low scores in other cell populations. These findings are now indicated in the text (p. 13, line 293) and shown in Suppl Fig. 6d.

- Related to this, it would be good to get deeper mechanistic insights into the link between AR and EGFR and TGF- β , potentially further analysis such as inference of gene regulatory network of the scRNA-seq data from the PDX could help or alternatively transcriptional analysis of the cell lines/mouse models treated with a BRAFi and AR inhibitors.

As suggested, we attempted to perform SCENIC analysis of the single cell PDX profiles reported by Rambow et al., Cell, 2018. Unfortunately, due to the sc-RNA seq resolution employed in this study, we were unable to detect AR transcripts at the individual cell level. In light of this limitation, we adopted an alternative experimental approach, as outlined in our general response above. This involved integrating transcriptomic and ChIP-seq analyses, along with functional assays employing inhibitors of EGFR and TGF- β signaling.

- Figure 3A, it would be very useful to name some of the relevant DE genes with high/low log2 fold change on the volcano plots.

We thank the reviewer for this comment. We have now added the relevant DE genes that are related to TGF β - and EGFR-signaling pathways (Fig. 5a).

Reviewer #2 (Remarks to the Author): expert in melanoma mouse models

*In this study, Samarkina and co-authors discovered that the Androgen Receptor (AR), besides being involved in cell proliferation and tumorigenesis, has an important role in BRAF inhibitor resistance of melanoma cells. MAPK-targeted therapies, such as BRAFi, have provided a clinical benefit not previously observed in metastatic melanoma of BRAF/RAS-mutated patients. However, despite all these efforts, melanoma treatment remains challenging because of the development of resistance to therapy. Here the authors demonstrate that AR increases upon BRAFi treatment and is upregulated in BRAFi-resistant melanoma cells. This is associated to the expression of EGFR/SERPINE1 genes both in melanoma cell lines and in publicly available datasets of melanoma patients. They further demonstrate that the pharmacological inhibition of AR in BRAF resistant cells suppresses tumor growth and induces CD8 T-cells infiltration/MHCI expression of BRAFi resistant cells-derived tumors in xenografts/syngeneic mouse models. Despite being very interesting and of high significance in the field, it is still quite preliminary and further data are needed to implement it. My major concern relies on the fact that the molecular mechanism(s) underlying the AR-mediated resistance phenotype in melanoma cells was not deeply investigated. An extensive bioinformatic analysis (both in melanoma cells and public datasets) has been done to identify the functional pathways/gene signatures modulated by AR OE and in BRAFi-resistant cells, but none of them has been experimentally evaluated. Another concern is the *in vivo* studies, which are quite limited. Experiments demonstrating that AR is upregulated also *in vivo* upon exposure to BRAFi and that tumor-bearing mice become resistant to BRAFi upon AR OE should be addressed.*

We appreciate the positive feedback provided by the reviewer regarding our work. In response to the queries concerning the molecular mechanism(s) underlying the AR-mediated resistance phenotype and the responsible pathways, we have thoroughly addressed this issue as explained in greater detail in our comprehensive response to reviewer #1. Briefly, we conducted a combined transcriptomic and ChIP-seq analysis using anti-AR antibodies of A375 melanoma cells plus/minus AR overexpression and compared these profiles with those of the same cells subjected to short-term BRAFi (DAB) treatment. Employing this approach, we successfully identified a specific set of genes that are selectively bound and upregulated by AR in cells with sustained expression of the AR gene, but not in cells with short term upregulation (by DAB treatment).

As we point out in the discussion (p. 19, line 450), the reasons for selective targeting of these genes can be the result of a complex cascade of events triggered by long-term AR upregulation and/or expansion of a pre-existing subpopulation of cells with differences in chromatin configuration and accessibility that affect the AR response. Irrespectively, we have found that these direct AR target genes are enriched for families related to TGF β - and EGFR-signaling pathways, both of which we now show to be involved in mediating AR-induced BRAFi resistance.

As per the reviewer's request, we have expanded our *in vivo* studies, as described in detail further below.

Comments, point-by-point:

- A general concern is that the statistics of many qPCRs analyses is missing, as for Fig. 1A,E,F, Fig. 5A and suppl. Fig 1A-D.

The statistical analysis has now been provided.

- Moreover, in Fig. 1A, why is there not any protein band of AR at 1uM of treatment? While the RNA is already upregulated at 0.5uM doses?

This difference can be explained by the different sensitivity of the two techniques and the fact that, due to a positive feedback loop, the earlier increase in AR mRNA level can be a reflection of increased AR activation, preceding an increase in protein levels.

- Densitometry is required in Fig. 1B,E

Densitometric quantification has been provided for Fig. 1B. In the revised version, Fig. 1E has been removed and replaced with Fig. 2b, where densitometry values are also included.

- The statistics is missing from the heatmap shown in Fig. 1D

The statistical analysis has now been provided.

- With a closer look at the original values of the first 3 genes from suppl. Fig 2A, mentioned in the text, they are actually pretty similar (DMSO/DAB). A qPCR should be done at least for the 3 genes in the different melanoma cell lines upon DAB treatment. Furthermore, a more in-depth examination of the role of these factors in AR regulation in BRAFi resistant/parental cells is warranted.

We tested by gene silencing whether the transcription factors identified by bioinformatic analysis as putative regulators of AR expression played indeed such a role. We obtained negative results and decided to omit this part of the study, given our other findings that AR itself is involved in induction of its own expression upon BRAFi treatment.

- The statistics is also missing in Fig. 2A.

The statistical analysis has now been provided.

- Why does Fig. 2E show a higher number of colonies in DMSO/CTR conditions, compared to Fig. 2D where the same A375 cells were used? The difference with AR OE is indeed quite low (compared to Fig. 2D).

The experiments presented in Figs. 4E and 4D were conducted independently, and the differences observed in the absolute number of colonies are due to inherent experimental and/or biological variability. To ensure the robustness and reliability of our findings, we always performed parallel controls for each experiment. Thus, despite the variations in absolute numbers, our data consistently demonstrate significant differences between DAB-treated cells plus/minus concomitant AR overexpression.

- It should be better clarified in the text lines 161, 164, in relation to Fig.3 C-E (now Fig. 5C-E). It is not so obvious from the text that the second GSEA plot (in C,D, and E) compares DAB ctr versus DAB AR OE.

For the sake of clarity, we have modified the way we show the data and refer to them in the text. Specifically, we now show in separate panels that, by Gene Set Enrichment Analysis (GSEA), signatures of interferon α - and γ response are highly induced by DAB treatment of control cells (Fig. 5C), with a strong difference in the DAB-treated control versus AR overexpressing cells (Fig. 5d). Similarly, gene signature encompassing many major histocompatibility class I (MHC I) genes are highly enriched in the DAB-treated control cells, with a profound difference relative to AR overexpressing cells (Fig. 5e,f).

- *Fig. 5B should include the densitometry analysis.*

The densitometry has now been provided.

- *The in vitro experiments in Fig.5 should be repeated upon genetic inhibition of AR, at least in some of the cell lines. All the human BRAFi-resistant cell lines used in the cell viability/cell death tests should be analysed for EGFR/SERPINE1 activation; more consistency with the different cell lines used in the different assays is required. In Fig. 5G, the untreated cells should also be included.*

We have provided the data on *EGFR/SERPINE1* expression levels upon shAR silencing in Suppl. Fig. 7d. The *EGFR/SERPINE1* expression levels for all BRAFi-resistant cell lines are now provided in Suppl. Fig. 6e-f.

- *The in vivo experiments in Fig. 6A-E should include the tumor kinetics analysis to see how the tumor growth of AZD/ARCC4 treated tumors decreases over time. Also, a more thoughtful analysis of the immune cell populations in the TME of YUMM1.7BR-derived tumors should be addressed. Flow cytometry analysis of at least T-cells populations should be included to confirm the IF data. Another weak point is that no characterization in terms of AR expression and cell viability upon BRAFi of YUMM1.7 BR cells has been done. It should be included to complete this part. Moreover, in both xenograft/syngeneic models further analyses showing the status of the BRAFi-resistant cells in terms of EGFR/SERPINE activation are missing.*

As indicated in the text (p. 16, top), the work with the syngeneic mouse model was extended by assessing the impact of systemic administration of AR inhibitors. Mice injected with the YUMM1.7BR cells were treated with AZD3514, ARCC4 and Enzalutamide by gavage, starting at day 3 after injection. As requested by the reviewer, we performed *tumor kinetics analysis* and show that tumour size expansion over time was significantly reduced in mice treated with all three AR inhibitors (Fig. 9f).

RNA extraction and RT-qPCR analysis of lesions at the end of the experiment showed that even *in vivo* AR inhibition resulted in the reduced expression of the *EGFR* and *SERPINE1* genes (Fig. 9c). Parallel IF analysis showed a striking decrease in Ki67 labeling of tumor cells in mice treated with the AR inhibitors versus controls (Fig. 9h).

Intra-tumoral localization of immune cells is a key determinant of the protective anti-tumor response³⁸. FACS analysis of dissociated cells from tumors at the end of the experiment revealed no significant changes in total numbers of CD8+ and CD4+ T cells

expressing various markers of functional activity (CD44, FOXP3) and exhaustion (PD-1, Tim-3, LAG-3). Rather, combined IF and 3D image reconstruction analysis showed a consistent and marked increase of clusters of CD8⁺ T cells, with interspersed CD4⁺ in multiple tumors of mice with systemic drug administration of AR inhibitors (Fig. 9i, Suppl. Fig. 8a).

The clusters of CD8⁺ cells were strongly positive for granzyme B expression, a family of serine proteases and marker of cytotoxic T lymphocytes that mediate their killing effects, with a concomitant localized staining of caspase 3 positive tumor cells (Fig. 9k, Suppl. Fig. 8b). The results were quantified and extended to tumors formed by YUMM1.7BR cells pretreated with AR inhibitors, showing even in this case, increased clustering of CD8⁺ T cells (Fig. 9l, m, Suppl. Fig. 8d-e).

As requested by the reviewer, we have also characterized the response of BRAFi resistant mouse melanoma cells (YUMM1.7BR) to AR inhibitors in culture finding it to be similar to that of human cells. Specifically, RT-qPCR and immunoblot analysis showed that, even for the YUMM1.7 cells, acquisition of BRAFi resistance was accompanied by the concomitant up-regulation of the *AR*, *EGFR* and *SERPINE1* (PAI-1) expression (Fig. 9a, b). Expression of all these genes was suppressed by treatment of the BRAFi resistant cells with the AR inhibitors (Fig. 9c) which, at the same time, suppressed their proliferation.

- In addition, if EGFR/TGF β pathways are hyperactivated once AR increases upon BRAFi exposure, can they be somehow targeted to rescue the AR-mediated resistance phenotype?

We thank the reviewer for the important question. As indicated in the text (p. 12, line 273), to assess whether increased TGF- β and/or EGFR signalling can account for the DAB resistance resulting from persistently increased AR expression, cells were treated with inhibitors of the two pathways, individually and in combination. Real time imaging assays showed that proliferation of A375 parental cells was unaffected by treatments with TGF- β and/or EGFR inhibitors, which also did not enhance the suppressive effects of DAB treatment (Fig. 6i). Proliferation of AR-overexpressing cells was similarly unaffected by treatment with TGF- β and/or EGFR inhibitors. However, the sustained proliferation of these cells in the presence of DAB was significantly reduced by the concomitant administration of either TGF- β or EGFR inhibitors, with stronger suppressive effects when the two were combined (Fig. 6i).

Reviewer #3 (Remarks to the Author): expert in BRAF inhibitors in melanoma

This study reports that the Androgen Receptor (AR) is a mediator of resistance to BRAF inhibitors (BRAFi) in BRAF mutant melanoma. It is shown that cells grown in high concentrations of BRAFi consistently upregulate AR, but not other common melanoma resistance genes. AR is upregulated within 48 hours of BRAFi or MEKi but is not upregulated by inhibitors of several other pathways. Tumours from patients who developed resistant also upregulate AR, and bioinformatic analysis shows that transcription factors (TF) known to upregulate AR are also increased, although the role of these TF is not tested directly. Cell growth when AR is over-expressed is less sensitive to BRAFi and the cells are more resistant to apoptosis. Bioinformatic analysis reveals that within 48 hours of BRAFi, pathways that control cell cycle are downregulated, but interferon and antigen presentation are upregulated. In cells over-expressing AR, Serpine and EGFR pathway components are upregulated, and these pathways correlate to distinct resistant sub-populations. High expression of AR/EGFR are associated with reduced survival. EGFR and Serpine are upregulated in cells when AR is over-expressed or the cells are grown to be resistant to BRAFi and the growth of the resistant cells is sensitive to AR antagonists. AR antagonists suppress the emergence of BRAFi colonies in vitro. Finally, pre-treatment of cells with AR antagonists suppresses human xenograft tumour growth and increased CD8+ T cell infiltration in BRAF mutant mouse allografts.

The data support the conclusions, but several weaknesses dampen enthusiasm for publication.

General comment. There is a lack of consistency between the use of cell lines and techniques and no single cell line is carried as an exemplar throughout all the study to give consistency of narrative and approaches. This makes it difficult to compare between cell lines and experiments, resulting in a broken narrative and making it difficult to be fully convinced by the data.

Throughout the study, we aimed to strengthen the validity of our observations by testing a number of established cell lines as well as freshly derived primary melanoma cells. Specifically, for key transcriptomic analysis and functional assays of increased AR expression, selection of BRAFi resistance and treatment with BRAFi plus/minus AR inhibitors, the same three melanoma cell lines were always included. To gain greater mechanistic insights, we consistently utilized A375 melanoma cells, which were always included in all other assays, except for experiments shown in Suppl. Fig. 2a and Fig. 8e. These cells were also used for the tumorigenicity assays in immune compromised mice (Fig. 8f-g) with a similarly characterized response being determined for the mouse BRAFi-resistant melanoma cells tested in syngeneic animals.

The same experimental approaches and techniques were used for analysis of the various cell lines, as specified in the figure legends and methods section.

Specifically:

1. Figure 1. The premise of this figure is that cells grown long-term in high concentrations of BRAFi (dabrafenib/DAB) become resistant to the inhibitor and increase AR expression. However, the resistant cells are not characterised to provide the reader with a baseline from which to understand the data.

a. *Cell growth against DAB concentration curves should be provided.*

We have now provided cell growth against DAB concentration curves in Supplementary Fig. 1D-H.

b. ERK activity against DAB concentration curves should be provided.

We have now provided results of ERK and phosphor-ERK immunoblotting in Supplementary Fig. 1N.

Panel E. It is curious that AR levels rise so quickly (within 48 hours), because most of these cells in these cultures will eventually die, which suggests that the AR elevation is insufficient to save most of the cells in the culture. The authors should test if the increase in AR in this time frame is restricted to a small population of cells that will survive and emerge as resistant. If all the cells express AR, how do the authors consolidate that observation with their model?

We thank the reviewers for the excellent questions, which served as guide for the further work and mechanistic insights that we have obtained. Our rationale stemmed from the reviewer's observation that short-term exposure of melanoma cells to BRAF inhibitors induces a notable increase in AR (androgen receptor) expression. However, this induction alone is insufficient to cause resistance to BRAF inhibitors, as it results instead from sustained AR expression.

As indicated in the text (p. 11, top), to investigate the underlying reasons, we performed ChIP-seq analysis using anti-AR antibodies on A375 melanoma cells plus/minus AR overexpression and compared these ChIP-seq profiles with those of AR in the same cells after early treatment with the BRAF inhibitor (DAB). The analysis revealed a significant increase in AR binding near the transcription start sites (TSS) of several genes in both AR-overexpressing A375 melanoma cells and the cells treated with DAB for 48 hours. However, there was only partial overlap between the AR target genes that were upregulated under these two conditions. Specifically, we identified one group of genes (65) that exhibited AR binding and induction in both AR-overexpressing cells and DAB-treated cells, and another group (202) that showed selective AR binding and induction solely in the AR-overexpressing cells (Fig. 6e).

Further analysis using gene set enrichment analysis (GSEA) revealed a significant enrichment of target genes related to TGF- β and EGFR signaling among the genes upregulated in AR-overexpressing cells (Fig. 6h). This is of functional significance as targeting these pathways effectively suppressed AR-induced BRAFi resistance (Fig. 6i).

We have also assessed whether the increase in AR expression upon short term exposure to BRAFi is restricted to a small population of cells. Our immunofluorescence analysis demonstrated the induction of AR expression in the majority of melanoma cells during the early stages of DAB treatment (Fig. 2f). This observation does not rule out the possibility of pre-existing subpopulations with heightened AR expression. Additionally, as considered in the discussion (page 19, line 448), it raises the intriguing possibility of subpopulations exhibiting distinct chromatin configurations that respond differentially to increased AR expression. This may provide an explanation for the observed differences in AR chromatin binding profiles and the associated alterations in gene expression between melanoma cells subjected to short-term DAB treatment and those with sustained elevation of AR expression.

d. *Please comment on the high concentrations of DAB used to drive resistance and how the concentration relates to the sensitivity of the parental lines.*

The stepwise increase in DAB concentrations used to select resistant melanoma cells is the same as previously reported in a number of studies to be effectively required for this to happen (Caporali et al., J Exp Clin Cancer Res, 2019; Yang et al., Nat Comm, 2021).

2. *Figure 2. Please provide western blots to confirm that AR over-expression compared to the parental cells and to the cells driven to resistance by growth in increasing concentrations of DAB.*

These data have now been provided in Suppl. Fig 4.

3. *Whilst Figures 3 and 4 present some interesting bioinformatic data, they do not carry the narrative much further, as largely the observations are not followed up. Most of this data could go to supplementary with just the key observations being presented in a reduced Figure 3.*

As explained below, we believe that the bioinformatic data presented in Figs. 5 and 7 (previously Fig. 3 and 4) provide crucial insights and molecular explanations for the functional findings. Moreover, due to the additional extensive analysis we have conducted, based on integrated transcriptomic and ChIP-seq profiling, we cannot consolidate the two figures without compromising the clarity and depth of the additional mechanistic insights that we have obtained on AR-induced BRAFi-resistance.

Regarding Fig. 5 (previously Fig. 3), it demonstrates that persistent AR expression alters the transcriptional response of melanoma cells to DAB treatment. Specifically, the downmodulation of gene families related to cell cycle and DNA replication in response to DAB treatment, which was observed in both control and AR-overexpressing melanoma cells, validates the decreased rate of proliferation observed in AR-overexpressing cells at early times of DAB exposure (Fig. 4, Fig. 5b, Suppl. Table 2). In contrast, the much greater upregulation of mitochondrial pro-apoptotic genes in control than AR-overexpressing cells upon DAB treatment provides a molecular underpinning for the differential pro-apoptotic effects of the drug (Fig. 5b, Suppl. Table 2). Additionally, the enhanced expression of genes in the EGFR and TGF- β pathways in AR-overexpressing cells (Fig. 5b, Suppl. Table 2) is of importance for the subsequent transcriptional (Fig. 6g-h) and functional (Fig. 6I) studies that we have performed. The results of the GSEA (Figure 5C-E), which reveal a differential impact on immunomodulatory genes by DAB treatment of control versus AR-overexpressing melanoma cells are significant for the clinical implications of the work, as we consider in the discussion (p. 18, bottom paragraph).

Regarding Fig. 7 (previous Fig. 4), this has now been significantly changed, with the addition of several new panels illustrating the comparative transcriptomic and ChIP-seq analysis discussed above.

4. *Figure 5.a. Panel C. Please provide western blots to accompany the RT-PCR data.*

The Western blots have now been provided to accompany the RT-qPCR data (Suppl. Fig.7C).

b. Panels E-G. Please demonstrate specificity by comparing the impact of AR antagonists on the growth of non-resistant parental cells.

As overviewed in the introduction, in our previous work (Ma et al., JEM 2021) we demonstrated the growth inhibitory effects of AR inhibitors on naïve melanoma cells. The purpose of the present experiments presented in Fig. 8c-e (previously Panels E-G of figure 5) is to show that BRAFi-resistant melanoma cells, in spite of the increased AR expression (as shown in Fig. 1a) still respond to AR inhibitors.

The narrative would be significantly strengthened if it was shown that EGFR and Serpine modulated resistance to BRAFi downstream of AR (i.e. are the cells addicted to these proteins?).

We thank the reviewer for the excellent suggestion. *SERPINE1* is part of the TGF β response genes that we found to be up-regulated in melanoma cells with AR overexpression over or selected for DAB-resistance (Fig. 6a). To assess whether increased TGF- β and/or EGFR signalling can account for DAB resistance, cells were treated with inhibitors of the two pathways, individually and in combination. Real-time imaging assays showed that proliferation of parental cells was unaffected by treatments with TGF- β and/or EGFR inhibitors, which also did not enhance the suppressive effects of DAB treatment (Fig. 6i, left panel). Proliferation of AR-overexpressing cells was similarly unaffected by treatment with TGF- β and/or EGFR inhibitors. However, the sustained proliferation of these cells in the presence of DAB was significantly reduced by the concomitant administration of either TGF- β or EGFR inhibitors, with stronger suppressive effects when the two were combined (Fig. 6i, right panel).

5. Figure 6

a. While the demonstration that pre-treatment of cells with AR antagonists reduces tumour growth in mice is of some interest, it is not surprising or particularly informative. The authors show in Figure 5F that AR antagonists cause cell death in vitro, and it is unsurprising that dying cells are less able to form tumours than healthy cell controls. The appropriate experiments to provide pre-clinical context are:

i. Demonstrate that AR antagonists suppress the growth of BRAFi resistant or AR-over-expressing BRAF mutant cells growing as established tumours in mice.
ii. Demonstrate that AR antagonists cooperate with BRAFi to suppress the emergence of resistant clones in parental cells growing as established tumours in mice.

b. Panels C-F. The functional relevance of MHC I surface expression and increased CD8+ T cell infiltration is unclear and untested. It could simply reflect the fact that the cells are dying when injected, and it is the cell death rather than AR antagonist specific effects that are causing the infiltration. The appropriate experiments are:

i. Provide (brief) evidence that YUMM1.7BR cells behave in the same way as human cell lines in vitro:

1. Is AR elevated in YUMM1.7BR cells compared to sensitive parental YUMM1.7 cells?
2. Are YUMM1.7BR cells sensitive to AR antagonists in vitro compared to the parental cells?

ii. Show that the increase of the immune cell infiltration occurs when mice bearing

established tumours are treated with AR antagonists.
iii. *Demonstrate specificity by showing T cell infiltration does not occur when BRAFi sensitive parental YUMM1.7 cells are treated with AR antagonists in mice.*

We thank the reviewer for the important questions regarding Fig. 6, now Fig. 9. We have addressed them as follows:

a) As per this request, we have extended our work by examining the impact of systemic administration of AR inhibitors on melanomagenesis. To achieve this, we treated mice injected with the YUMM1.7-BR cells with three different AR inhibitors, namely AZD3514, ARCC4, and Enzalutamide, via gavage. The treatment was initiated three days after injection, when the tumor's incipient outgrowth occurs. Our results show that the expansion of tumor size over time was significantly reduced in mice treated with all three AR inhibitors (Fig. 9f). We have tested the impact of AR inhibitors on the emergence of BRAFi-resistant clones under well-defined culture conditions. To address this question by serial dilution injections of cells *in vivo* would require a large number of mice and the clinical significance of the findings is now also supported by a set of clinical and mouse models data published while our paper was under review (Vellano et al., Nature, 2022), to which we refer in both introduction and discussion (p. 4, line 84 and p. 22, line 480).

b) as requested, we show that the mouse melanoma cells selected for BRAFi resistance (YUMM1.7-BR) exhibit, relative to the parental cell line, a significant up-regulation of the *AR*, *EGFR* and *SERPINE1* (PAI-1) genes (Fig. 9a) and that expression of all these genes was suppressed by treatment of these cells with the AR inhibitors (Fig. 9c) which, at the same time, suppressed their proliferation (Fig. 9d).

c) By RNA extraction and RT-qPCR analysis of the lesions at the end of the experiment we show that even *in vivo* administration of AR inhibitors by gavage resulted in reduced expression of the *EGFR* and *SERPINE1* genes (Fig. 9g).

d) As requested, we have removed the IF analysis of MHC expression and have focused on the proliferation and apoptosis of cancer cells, by staining with anti-Ki67 and cleaved caspase 3 antibodies, respectively, and on the infiltration of immune cells (Fig. 9h-m, Suppl. Fig. 8).

FACS analysis of dissociated cells from tumors at the end of the experiment revealed no significant changes in total numbers of CD8⁺ and CD4⁺ T cells expressing various markers of functional activity (as shown here below for the reviewer's consideration). As an alternative approach, we resorted to IF analysis combined with thick sectioning of lesions and 3D image reconstruction, which showed a consistent and marked increase of clusters of CD8⁺ T cells, with interspersed CD4⁺ in multiple tumors of mice with systemic drug administration of AR inhibitors (Fig. 9i, Suppl. Fig. 8a).

The clusters of CD8⁺ cells were strongly positive for granzyme B expression, a family of serine proteases and marker of cytotoxic T lymphocytes that mediate their killing effects, with concomitant localized staining of caspase 3 positive tumor cells (Fig. 9k, Suppl. Fig. 8b). The results were quantified and extended to tumors formed by YUMM1.7BR cells pretreated with AR inhibitors, showing even in this case, increased clustering of CD8⁺ T cells (Fig. 9l,m, Suppl. Fig. 8c-e).

Regarding the impact of AR inhibitors on immune cells infiltration of tumors formed in a syngeneic mouse model by the parental mouse melanoma cells, this was investigated in our previous publication, in which no net increase in infiltrating T cells was also observed (Ma et al., JEM 2021).

Reviewer #4 (Remarks to the Author): expert in androgen receptor signalling

This manuscript by Samarkina et al., is focused on unravelling a novel mechanism of resistance in response to BRAF/MEK inhibitors in melanoma patients. In this study, the authors have shown that expression of androgen receptor (AR) is significantly elevated in melanoma cells exposed to BRAF/MEK inhibitors. Increased AR expression is important for maintaining the tumorigenicity in these cells and is accompanied with high levels EGFR and SERPINE1 expression which could confer a tumor survival advantage. In addition, blocking AR signaling in BRAFi-resistant cells leads to increased expression MHC I expression and CD8+ T cells infiltration. This body of work is very important in setting a possible precedent for androgen receptor blockade as a therapy against BRAFi/MEKi-resistance in melanoma patients. However, this study seems more correlative in nature and does not dwell deeper into the mechanism behind AR upregulation. Moreover, this study relies much on cell line data rather than clinical melanoma data. Attention to the following points can further help improve the quality of the manuscript.

We thank the reviewer for the positive evaluation of the work. We have made significant efforts to enhance the mechanistic insights as explained in greater detail in reply to reviewer 1. Briefly, we conducted a combined transcriptomic and ChIP-seq analysis using anti-AR antibodies of A375 melanoma cells with or without AR overexpression and compared these profiles with those of the same cells at early times of BRAFi treatment, which results in an acute induction of AR expression not sufficient to cause BRAFi resistance.

We have identified a specific set of target genes that are selectively bound and upregulated by AR in cells with sustained expression of this gene but not in cells with short-term upregulation. We have found that these direct AR target genes are enriched for families related to TGF β - and EGFR-signaling pathways, which we now show by pharmacological inhibition studies to be involved in mediating AR-induced BRAFi resistance.

We acknowledge in the discussion that the selective targeting of these genes may be the result of a complex cascade of events triggered by long-term AR upregulation and/or expansion of a pre-existing subpopulation of cells with differences in chromatin configuration and accessibility that affect the AR response.

In terms of clinical significance, we have assessed AR expression by IF analysis of matched lesions excised from two patients before and after BRAFi/MEKi therapy. Analysis of topographically distinct areas for levels of single-cell AR expression revealed a consistent upregulation of AR expression throughout the lesions arising after BRAFi treatment (Fig. 1H). The functional assays were validated by testing freshly derived primary melanoma cells in parallel with melanoma cell lines and in a syngeneic mouse model (Fig. 9).

Finally, in both introduction (p. 4, line 84) and discussion (p. 20, line 480), we refer to a comprehensive series of clinical and mouse model data published during the review process of our paper (Vellano et al. 2022), which reinforces the overall importance of our findings.

Major Comments:

1. The author should include the recent study by Vellano et al., Nature, 2022 in the introduction.

As indicated above, we now consider Vellano's paper in both introduction (p. 4, line 84) and discussion (p. 20, line 480).

2. In several bar graphs, error bars are missing. Please add the error bars.

We have added the error bars as requested.

3. Some of the figure legends are overly descriptive. The figure legends should be more succinct, and the information about the experiment details should be explained in the methods section.

We thank the reviewer for this comment. We have moved information regarding experimental details to the methods section and made the figure legends more succinct.

4. *Figure 1, apart from checking AR expression, the authors should also check the expression of AR-regulated genes from prostate cancer. This will also help in understanding the distinct mechanism of regulation mediated by AR in melanoma.*

We thank the reviewer for the suggestion. In Fig. 1f, we assessed levels of AR activity in various BRAFi-resistant melanoma lines by GSEA of transcriptomic profiles using a hallmark AR-responsive gene signature derived mostly from prostate-derived profiles showing a highly significant enrichment.

For a wider perspective, as indicated in the text (p. 20, line 471), we used 17 different gene signatures measuring AR activity in prostate cells derived from the literature for GSEA of our profiles of melanoma cells under control conditions versus selected for DAB resistance. Leading-edge analysis of GSEA plots showed a significant number of genes that are common to several prostate-derived gene signatures and concordantly modulated as a function of BRAFi resistance (Suppl. Fig. 9b and Suppl. Table 1).

5. The authors should perform some experiments with androgens to confirm the ligand-dependent or ligand-independent activity of AR in melanoma cells.

We thank the reviewer for the question. We now show that, in contrast to the inhibitory effects of AR inhibitors on the growth of BRAFi-resistant melanoma cells, the treatment with the AR agonist dihydrotestosterone (DHT) significantly increased their proliferation (Suppl. Fig. 7e).

6. In Supplementary Figure 1D, protein level for AR in other melanoma cell lines would strengthen the manuscript.

We are now showing in Fig. 2b-d and in Suppl Fig. 2b AR protein levels at various time points in multiple melanoma cell lines.

7. The authors have established the connection of FOXO3, CREB1 and SP1 with AR expression based on the literature and correlative expression patterns. The authors should also perform some genetic perturbations in either parental cells (overexpression) or BRAF resistant cells (knockdown) to confirm whether these proteins are regulating AR expression.

As recommended, we have tested by gene silencing whether the transcription factors identified by bioinformatic analysis as putative regulators of AR expression played indeed such a role. We obtained negative results and decided to omit this part of the study, given our other findings that AR itself is involved in induction of its own expression upon BRAFi treatment.

8. In Figure 2, author should perform additional functional assays (such as invasion, migration or tumor sphere formation assays) to assess the role of AR overexpression in melanoma cells in response to BRAF inhibition.

We have performed tumor spheroid invasion assays as requested. AR overexpression resulted in increased invasion capability of cells under basal conditions and their migration out of spheroids remained higher than controls upon DAB treatment (Suppl. Fig. 5).

9. In Figure 3, the authors should also perform GSEA analysis for androgen signaling to check for the expression of conventional androgen-regulated genes.

As requested, and indicated in reply to point 4 above, we performed GSEA of transcriptomic profiles of various BRAFi-resistant melanoma lines using a hallmark signature of AR-responsive genes derived mostly from prostate-derived profiles (Fig. 1f). We also used multiple distinct gene signatures of AR activity in prostate cells for GSEA of our profiles. Leading-edge analysis of GSEA plots showed a significant number of genes

that are common to several prostate-derived gene signatures and concordantly modulated as a function of BRAFi resistance (Suppl. Fig. 9b, listed in Suppl. Table 1).

10. In supplementary figure 4, authors have used the AR inhibitors AZD3514 (10 μ M) or ARCC4 (1 μ M). Is there any particular reason why authors did not use anti-androgens such as enzalutamide or apalutamide, that are frequently used to inhibit AR activity in prostate cancer?

In our hands, for the present study as well as in our previous work (Ma et al., JEM 2021), AZD3514 (10 μ M) and ARCC4 (1 μ M), which function by affecting AR protein levels are more active than enzalutamide. For the sake of completeness, the latter inhibitor was also added to the *in vivo* study based on gavage administration of AR inhibitors obtaining similar, if slightly weaker effects (Fig. 9f, g).

Minor Comments:

1. In Figure panel 1D and 3B, the authors should consider different colours for the heat map to distinguish between the high and low gene expression changes.

We thank reviewer for this comment. We incorporated the changes accordingly (Fig. 3b is now Fig. 5b).

REVIEWERS' COMMENTS

Reviewer #1 (Remarks to the Author):

The authors have addressed my comments. The manuscript has been considerably improved with the addition of new studies.

Reviewer #2 (Remarks to the Author):

The authors have addressed most of the concerns, and the revised manuscript is highly improved.

The text should be revised in relation to the figures indicated, as there are some mistakes (e.g., line 102,343,347,378).

Reviewer #3 (Remarks to the Author):

To address the Reviewers' concern that the data originally presented were correlative and lacked functional testing, the authors have added a good deal of new data. This does largely address the previous concerns by adding new insight into mechanism and the therapeutic potential of the approach. Overall the study is much improved, although as noted below, the response to treatment that results from the study is, at best, modest, and this weakens the final conclusions that can be drawn.

The following issues should be addressed.

Major.

1. Figure 9f. This is the crux of the study, as it shows that BRAFi resistant melanoma cells are sensitive to AR antagonists. While the data attracts highly significant p-values the responses are, at best, modest and it is notable that the tumors continue to grow rapidly and double from 500mm³ to 1000mm³ in just a few days for 2 of the 3 antagonists. In a patient this would be classed as progressive disease, so the authors should acknowledge this limitation and discuss whether this result would warrant clinical testing of this treatment in patients.

Minor.

1. Fig 2a. Sorafenib is not considered, by most, to be a BRAF inhibitor.

2. Fig 2f. The DAB samples need DAPI controls.

3. Fig 4b. It is impossible to distinguish the lines for the individual experiments. It may be better to show only one concentration of DAB, and either move the rest to the Supplemental Data or exclude them altogether.

4. Line 210. The use of "paradoxically" in this sentence could be confused by the use of this term to describe pathway hyperactivation by RAF inhibitors in other contexts. It is not clear if what is observed here is a true paradox, or just normal gene expression changes (some go up, some go down) for this type of analysis. The word "paradoxically" should be replaced.

5. The authors may wish to make the point that this pathway is active in both male and female melanomas, as both male and female cell lines were analysed, and both male and female patient data were analysed.

Reviewer #4 (Remarks to the Author):

The authors have been responsive and satisfactorily address this reviewer's comments.

Reply to reviewer 3.

We thank the reviewer for the favorable evaluation and have addressed the remaining concerns as follows :

Major.

1. Figure 9f. This is the crux of the study, as it shows that BRAFi resistant melanoma cells are sensitive to AR antagonists. While the data attracts highly significant p-values the responses are, at best, modest and it is notable that the tumors continue to grow rapidly and double from 500mm³ to 1000mm³ in just a few days for 2 of the 3 antagonists. In a patient this would be classed as progressive disease, so the authors should acknowledge this limitation and discuss whether this result would warrant clinical testing of this treatment in patients.

As we had pointed out at the end of the introduction (p. 5, 6th line from the top), the clinical relevance of the findings is also supported by " a comprehensive series of clinical and mouse model data published while this paper was under review ¹⁵".

The notion that AR targeting is of potential clinical significance as a co-adjuvant of other therapeutic efforts is further considered at the end of the discussion, where we point out that (p. 16, 4th line from top) : "as also indicated by another study ¹⁵, inhibitors targeting AR activity and expression could therefore be employed as co-adjuvants to prevent/delay targeted drug resistance and, as we have shown, reduce (a term which we have used to replace "suppress" in response to the reviewer's concern) tumorigenicity of BRAFi-resistant cells while at the same time enhancing CD8⁺ T cells infiltration. Given the connection between BRAFi resistance and poor immune response ¹³, as well as the intrinsic role of AR activation in dampening the T cell activity ^{12,47}, AR targeting may be beneficial in the treatment regimens with immune checkpoint inhibitors".

We further note that the experiment shown in Fig. 9f involves the treatment of mice with AR inhibitors 3 days after tumor cells injection. To better reflect the finding, as the reviewer suggests, we now state that "tumor size expansion was significantly reduced at later times in mice treated with all three AR inhibitors (Fig. 9f) (p. 12, 5th line from bottom). There may be multiple reasons why no effects were observed at earlier times, including the build-up of critical concentrations of inhibitors administered by daily gavage.

Minor.

1. Fig 2a. Sorafenib is not considered, by most, to be a BRAF inhibitor.

We thank the reviewer for pointing out that Sorafenib is a multikinase inhibitor, which has B-RAF and other kinases of the Ras-MAPK pathway as some of its targets. We now specify this in the figure legend (p.33, 3rd line from top).

2. Fig 2f. The DAB samples need DAPI controls.

We have added the DAPI images

3. Fig 4b. It is impossible to distinguish the lines for the individual experiments. It may be better to show only one concentration of DAB, and either move the rest to the Supplemental Data or exclude them altogether.

Fig. 4b shows the result of live-cell imaging analysis (IncuCyte) of the proliferation of melanoma cells obtained as in panel (a) cultured with only one concentration of Dabrafenib as indicated versus DMSO control. Cells were plated / tested on multiple wells per condition (n=3) and the standard bar, as it is typical for this kind of assays, refers to the range of values per condition per time point.

4. Line 210. The use of "paradoxically" in this sentence could be confused by the use of this term to describe pathway hyperactivated on by RAF inhibitors in other contexts. It is not clear if what is observed here is a true paradox, or just normal gene expression changes (some go up, some go down) for this type of analysis. The word Page 4 of 4 "paradoxically" should be replaced.

The word "paradoxically", which is not necessary, was removed (p. 8, 15th line from top).

5. The authors may wish to make the point that this pathway is active in both male and female melanomas, as both male and female cell lines were analyzed, and both male and female patient data were analyzed.

We thank the reviewer for making this important point, which we now make at the beginning of the discussion (p. 13, 8th line from bottom).